# Mesenchymal Stem Cell Extract Promotes Skin Wound Healing

**DOI:** 10.3390/ijms252413745

**Published:** 2024-12-23

**Authors:** Zi Deng, Kengo Iwasaki, Yihao Peng, Yoshitomo Honda

**Affiliations:** 1Department of Oral Anatomy, Osaka Dental University, Osaka 573-1121, Japan; dengzi1014@gmail.com (Z.D.); honda-y@cc.osaka-dent.ac.jp (Y.H.); 2Advanced Medicine Research Center, Translational Research Institute for Medical Innovation (TRIMI), Osaka Dental University, Osaka 573-1121, Japan; 3Department of Periodontology, Osaka Dental University, Osaka 573-1121, Japan; pengyh910806@outlook.com

**Keywords:** mesenchymal stem cells, cell extract, proliferation, wound healing, therapeutic potential

## Abstract

Recently, it has been reported that mesenchymal stem cell (MSC)-derived humoral factors promote skin wound healing. As these humoral factors are transiently stored in cytoplasm, we collected them as part of the cell extracts from MSCs (MSC-ext). This study aimed to investigate the effects of MSC-ext on skin wound healing. We examined the effects of MSC-ext on cell proliferation and migration. Additionally, the effect of MSC-ext on skin wound healing was evaluated using a mouse skin defect model. The MSC-ext enhanced the proliferation of dermal fibroblasts, epithelial cells, and endothelial cells. It also increased the number of migrating fibroblasts and epithelial cells. The skin defects treated with MSC-ext demonstrated rapid wound closure compared to those treated with phosphate-buffered saline. The MSC-ext group exhibited a thicker dermis, larger Picrosirius red-positive areas, and a higher number of Ki67-positive cells. Our results indicate that MSC-ext promotes the proliferation and/or migration of fibroblasts, epithelial cells, and endothelial cells, and enhances skin wound healing. This suggests the therapeutic potential of MSC-ext in treating skin defects as a novel cell-free treatment modality.

## 1. Introduction

The skin is the largest organ of the body. It protects internal organs from external damage, balances electrolytes, provides immune defense, aids in thermoregulation, maintains hydration, and senses external stimuli through sensory nerve endings [1]. Owing to its direct contact with the external environment, the skin is susceptible to various injuries, including cuts, burns, and infections. Although skin disorders are common and can affect people of all ages, poor skin wound healing can have serious consequences and is a major global public health issue [2,3,4]. Skin injuries are generally treated via wound cleansing, antibiotic ointments, protective dressings, and anti-inflammatory medications, but more complex treatments are required for severe or extensive wounds, including surgery, skin grafting, and long-term rehabilitation [5,6,7]. However, the problems of poor healing and scarring in persistent skin lesions remain, necessitating the development of novel therapeutic approaches [8,9].

Mesenchymal stem cells (MSCs) are a cell population originally isolated from bone marrow fluid in the 1970s by Friedenstein et al. as fibroblast-like cells that adhered to and proliferated on plastic surfaces [10]. MSCs are known to differentiate into bone, cartilage, and muscle, and possess functions that are beneficial for wound healing, such as immunomodulation, anti-inflammation, anti-apoptosis, fibrosis inhibition, and angiogenesis [11,12,13,14]. The application of these cells in regenerative medicine has been widely investigated [12,13,14,15,16]. The various functions of MSCs in improving wound healing depend on the humoral factors they produce, suggesting the potential for regenerative medicine using MSC-derived culture supernatants and exosomes [17,18,19,20,21,22,23]. It has been suggested that MSC-derived humoral factors may contribute to tissue regeneration by enhancing the proliferation and migration of the cells suppressed under low oxygen and low nutrient conditions [24]. Regenerative therapies using humoral factors can overcome the limitations associated with MSC transplantation, such as the need to secure a stem cell source, the risk of tumorigenesis caused by the transplanted cells, the risk of spontaneous growth and the differentiation of the transplanted cells, and high treatment costs. Treatment with humoral factors is called cell-free treatment and is currently attracting attention as a new stem cell-based therapy [21,25,26,27].

MSC culture supernatants and MSC-derived exosomes are often utilized in cell-free treatment; however, the amount of humoral factors released from MSCs is small, and these factors are diluted in the cell culture medium during collection, necessitating a large amount of cell culture [28,29]. Previously, we focused on the fact that these humoral factors are transiently stored in the intracellular cytoplasm and hypothesized the possibility of disrupting the cell membrane to recover these factors as a cell extract. We recovered a cellular extract from the immortalized bone marrow-derived MSCs through freeze–thaw cycles and named this extract “MSC-ext”. We previously found that this MSC-ext enhanced the proliferation of periodontal ligament cells, which are important for the regeneration of tooth support tissue [30], and that the administration of MSC-ext induced periodontal tissue regeneration [31]. The MSC-exts prepared from the immortalized MSCs have advantages, such as being free from the effects of the cellular senescence of stem cells during MSC expansion and being able to be produced in large quantities at a low cost, and may be a new cell-free treatment method for tissue regeneration. However, their ability to promote wound healing in skin defects is not known. The purpose of this study is to investigate the effects of MSC-ext on fibroblasts, epithelial cells, and vascular endothelial cells, which play a role in skin wound healing, and to examine the effect of MSC-ext on wound healing using a skin defect model in mice.

## 2. Results

### 2.1. MSC-Ext Promotes the Proliferation of Dermal Fibroblasts, Epithelial Cells, and Vascular Endothelial Cells

The MSC-ext was prepared as shown in Figure 1. The average amount of MSC-ext collected from the 10^6^ UE7T-13 cells was 188 ± 15.6 µg. We examined the effects of MSC-ext on cell proliferation in vitro. The cell proliferation results for the dermal fibroblasts (NHDF), epithelial cells (HSC-1), and vascular endothelial cells (MS1) treated with MSC-ext are shown in Figure 2A–C, respectively. MSC-ext promoted NHDF proliferation at a protein concentration of 175 μg/mL. Cell proliferation was also enhanced when 1750 μg/mL of MSC-ext was applied to HSC-1 and MS1 (Figure 2B,C). In addition, no abnormal morphological changes were observed in the cells treated with MSC-ext as shown in Figure 2D. Similar results were also observed in immunofluorescence staining of the cells for Ki67, a proliferating cell antigen. As shown in Figure 2E,F, the addition of MSC-ext increased the percentage of Ki67-positive cells in NHDF, HSC-1, and MS1.

### 2.2. MSC-Ext Enhances the Migration of Dermal Fibroblasts and Epithelial Cells

Next, the effects of MSC-ext on cell migration in NHDF, HSC-1, and MS1 cells were examined using the Boyden chamber assay and wound healing assay. Figure 3A shows images of the transwell membrane at lower and higher magnifications. MSC-ext increased the area of crystal violet-positive staining in the NHDF and HSC-1 cells. A quantification of the migrating cell counts is shown in Figure 3B. The number of migrated cells significantly increased in the NHDF and HSC-1 cells after the MSC-ext application. In contrast, MSC-ext had no effect on cell migration in the MS1 cells (Figure 3A,B). In a wound healing assay, we observed that the NHDF and HSC-1 cells in the MSC-ext-treated group closed the scratch gaps more rapidly compared to the PBS-treated control group, whereas the MS1 cells showed no significant change. This finding aligns with the results of the transwell assay. The data showed that MSC-ext treatment significantly accelerated the migration rate of NHDF cells. After 18 hours of treatment, the migration rate of NHDF cells was approximately 40% higher than that of the PBS-treated control group. Similarly, MSC-ext treatment markedly enhanced the migration rate of HSC-1 cells, which was about 30% higher compared to the PBS-treated control group. In contrast, MSC-ext had no effect on cell migration in the MS1 cells (Figure 3C,D).

### 2.3. MSC-Ext Accelerates the Healing of Skin Defects

Dermal fibroblasts, epithelial cells, and vascular endothelial cells participate in skin wound healing. The results showed that MSC-ext promoted the proliferation and migration of these cells, suggesting that MSC-ext may improve skin wound healing. We examined the effects of MSC-ext on skin wound healing in a skin defect mouse model. Figure 4A illustrates a schematic representation of the method for administering MSC-ext to skin defects in mice. The MSC-ext was dissolved in PBS and was administered subcutaneously around the defect in four locations for five consecutive days, as the MSC-ext would flow when administered to the defect surface and local retention would be extremely low. The defects treated with MSC-ext were significantly smaller than those treated with PBS after day 4, indicating that MSC-ext accelerated skin defect healing (Figure 4B,C).

### 2.4. MSC-Ext Enhances the Thickness of the Regenerating Dermis and Promotes the Formation of Collagen Fibers in the Dermis

Healing tissue was collected on postoperative day 5, when the difference in defect closure between the PBS-treated control and MSC-ext groups was greater, and wound healing was compared histologically between the two groups. The dermal thickness was measured to assess granulation tissue formation, an important step in wound healing for epithelial layer development and connective tissue healing. Figure 5A shows the representative Azan staining images of the PBS and MSC-ext groups, showing a thickened dermis and enhanced collagen fiber formation in the MSC-ext group when compared to the PBS group. The quantifications of the epidermal and dermal thicknesses in the tissue specimens showed that the dermis was significantly thicker in the MSC-ext group than in the PBS-treated control group (Figure 5B); however, there was no difference in epidermal thickness (Figure 5B). A polarized observation of the Picrosirius red-stained sections showed a trend toward an enhanced polarized area in the MSC-ext group compared to the PBS-treated control group (Figure 5C), although there was wide variation among samples. In addition, the infiltration of many of the inflammatory cells was observed in the tissue; however, there was no statistically significant difference in the number of inflammatory cells in the dermal connective tissue (Figure 5D,E).

### 2.5. MSC-Ext Increases the Number of Ki67-Positive Cells in Dermal Connective Tissue

Figure 6A shows the results of the Ki67 immunofluorescence staining in the dermis; many Ki67-positive cells in the MSC-ext group were found in the fibroblasts between the collagen fibers. The results indicated that the MSC-ext-treated group had a significantly higher percentage of Ki67-positive cells than the PBS-treated group (Figure 6B).

## 3. Discussion

In this study, we found that MSC-ext promoted the proliferation of fibroblasts, epithelial cells, and endothelial cells, and the migration of fibroblasts and epithelial cells, suggesting that MSC-ext can promote the functions of the cells that have an important role in skin wound healing. These results inspired us to examine the therapeutic effect of MSC-ext on skin defect healing. We observed the accelerated healing of skin defects when MSC-ext was administered.

MSC-ext has significant positive effects on fibroblasts, epithelial cells, and endothelial cells in in vitro experiments. However, we could not identify the factors central to this phenomenon. Since MSC-ext is recovered from MSCs through freeze–thaw cycles, and thus may comprise a diverse array of factors derived from the MSCs. In a previous study, it was found that the application of protein fraction of MSC-ext to periodontal ligament cells significantly promoted their proliferation, primarily through accelerated cell cycle progression and increased mitotic activity [30]. These findings suggest that the proteins within the MSC-ext play a key role in enhancing cell proliferation. Moreover, we examined the protein components of MSC-ext in detail using liquid chromatograph–mass spectrometry (LC-MS/MS) and found that it contained 4388 proteins and was composed of many matrix proteins, enzymes, and cytoskeletal proteins. Additionally, using a protein array analysis it has also been revealed that MSC-ext is rich in some growth factors including the basic fibroblast growth factor (bFGF) and the hepatocyte growth factor (HGF) [31]. The bFGF is well known for its mitotic activity [32,33] and has been reported to enhance the proliferation of fibroblasts, epithelial cells, and vascular endothelial cells [34,35,36]. The HGF has been identified as a growth factor for liver cells with high regenerative activity, but also promotes cell proliferation in cells other than hepatocytes [37,38]. Furthermore, the bFGF and the HGF are known to act as cell migration factors, in addition to their proliferative activities [39,40]. Therefore, it is possible that the results of this study showing the positive effects on fibroblasts, epithelial cells, and endothelial cells were regulated by the bFGF and/or HGF contained within the MSC-ext. However, this point must be confirmed in future research using neutralizing antibodies.

In contrast, the effect of MSC-ext on endothelial cell migration was not observed, despite its promotion of fibroblast and epithelial cell migration. This discrepancy may be due to the complex interactions between the growth factors and the other components present in MSC-ext. The limitation of the endothelial cell migration experiment is the absence of a positive control group, which would provide a baseline for evaluating the effectiveness of the MSC-ext. While the bFGF and other factors significantly enhanced the migration of fibroblasts and epithelial cells, the vascular endothelial growth factor (VEGF) is a key chemotactic factor for endothelial cells [41]. The high content of the bFGF and the HGF in MSC-ext may have caused the fibroblasts and epithelial cells to migrate, while the migration of the vascular endothelial cells was not altered due to the low content of VEGF [31].

In further investigating the role of MSC-ext in skin wound healing, we conducted an in vivo experiment in which MSC-ext was locally injected into a skin defect mouse model. We observed an earlier closure of the skin defects in the MSC-ext-treated mice compared to the PBS-treated mice. Additionally, the number of Ki67-positive cells in the dermal layer increased, indicating enhanced cell proliferation in vivo. These findings suggest that MSC extracts promote wound healing. Mishra et al. reported that through ultrasonic disruption the cell lysates derived from bone marrow MSCs promoted skin defect closure in NOD-SCID mice [42]. While their experimental system differed from ours regarding the use of primary-cultured MSCs and nude mice, and the number of lysates administered, their findings align with ours. Furthermore, Na et al. reported that cell extracts prepared from adipose-derived MSCs shorten the duration of skin defect closure in mice [43]. Their study differs from the present study in that they used adipose-derived primary MSCs and only administered the cell extract once. However, their results are consistent with our results, similarly suggesting that MSC-derived cell extracts promote the healing of skin defects. Although previous reports and the present results indicate that MSC extracts promote wound healing in skin defects, the mechanism of such healing remains unclear. Na et al. [43] showed that adipose-derived MSC extracts promoted the proliferation and migration of dermal fibroblasts, increased the type 1 collagen and MMP1 production of fibroblasts, and considered this to be a possible mechanism of skin wound healing promoted by the extracts. Although these results are consistent with ours, we found that MSC-ext affects not only dermal fibroblasts but also epithelial cells and vascular endothelial cells. We speculate that the effects of MSC-ext on these cell types play an important role in skin wound healing and may be the mechanism by which MSC-ext promotes skin defect closure. The mechanism by which MSC-ext improves skin wound healing should be elucidated in further studies using healing tissues in vivo.

Since MSC-ext is an extract from cells, it is possible that it could have negative as well as positive effects when administered in in vivo experimental models. Inevitably, MSC-ext also contains nucleic acid components such as miRNA, which are closely associated with inflammation regulation. We compared the number of inflammatory cells in the MSC-ext-treated skin samples and found no significant difference in the infiltration of inflammatory cells in this group compared to the PBS group, suggesting that MSC-ext did not induce an excessive inflammatory response, which may be one mechanism of action that promoted skin wound healing. However, further investigation is required to explore the potential inflammation-related cellular changes and their underlying mechanisms in this study.

In this study, we found that the extracts from immortalized MSCs promote cell proliferation, migration, and skin wound healing. However, the safety of using extracts derived from immortalized MSCs for clinical therapy has not yet been verified. Further investigations are required to address various safety concerns, including the potential contamination of the extracts with immortalized cells, mutations in the protein components caused by the retroviruses used for immortalization, and contamination by unknown animal-derived components. Moreover, it is essential to verify whether the extracts from the primary-culture MSCs, which are considered safer, exhibit effects comparable to those from the immortalized MSCs. Previous studies have reported that the extracts obtained from umbilical cord, adipose tissue, and bone marrow MSCs promote anti-inflammation, immune regulation, cell proliferation, and tissue repair [44,45,46]. Since the production of many cellular factors is known to differ between immortalized cells and primary-cultured MSCs [47,48], future studies should evaluate the effects of primary-cultured MSC extracts on skin wound healing.

Although this study suggests a positive effect of MSC-ext on skin wound healing, it has several limitations. First, the skin defect model used in this study was created on the healthy back skin of young mice, which excluded the effects of chronic inflammation, severe infection, metabolic diseases, and the aging of patients, which would require additional treatment for persistent skin lesions in a clinical setting. Second, the healing period in this study was limited to 13 days, and the safety and efficacy of the treatment during the long-term follow-ups are unknown. Third, although the MSC-ext used in this study has been confirmed to exhibit cell proliferative effects, future studies should explore whether the wound healing effects of the MSC-ext in this study involved anti-inflammatory or immunomodulatory actions, and whether the extracts from primary MSC cultures have similar effects on skin wound healing to those from immortalized MSCs. More importantly, immortalized MSCs avoid aging through telomere maintenance and senescence prevention, but they are still vulnerable to cellular stress, DNA damage, and epigenetic changes [49]. While useful for research, caution is needed when applying these cells in clinical therapies due to potential tumorigenic risks. Furthermore, the concentration, frequency, and route of administration of MSC-ext used in vivo has not been fully optimized, and the search for highly effective administration methods remains challenging. The limitations of this study will be addressed in future experiments.

Recently, a variety of new treatments for skin injuries have been developed and reported. These include biomaterials with unique drug-elimination properties [50], inorganic biomaterials that accelerate wound healing [51], nano-materials [52], stem cell transplantation, and the transplantation of the liquid factors derived from stem cells. Although no carriers were used in this study, it is conceivable that the use of superior carriers for a sustained release of MSC-ext may increase its therapeutic efficacy, so future exploration or the development of suitable carriers for MSC-ext may be necessary.

MSCs are known to promote the proliferation and migration of various cell types [24]. Additionally, they exhibit properties such as anti-inflammatory effects, immunomodulatory effects, anti-apoptotic effects, and anti-fibrotic effects, which positively regulate wound healing [11,12,13,14]. Many clinical and preclinical studies have shown that MSC transplantation can improve wound healing and is a promising new therapeutic strategy for skin wounds [53,54]. However, the limitations and drawbacks of MSC therapy have been highlighted [55,56,57]. Surgical intervention is required for an MSC culture, and immunosuppressive treatment is necessary for allogeneic MSC transplantation. In addition, there is no strategy for controlling the spontaneous growth and differentiation of MSCs after transplantation, which raises the issue of treatment safety, including tumorigenesis. Furthermore, the treatment costs tend to be high as a result of the high cost of clinical application-grade cell cultures. MSCs do not undergo tumorigenic transformation and are generally considered safe under standard culture conditions [58]. However, the growth factors, cytokines, as well as chemokines secreted by MSCs may influence the tumor microenvironment by promoting the proliferation of cancer cells. Such effects, however, depend on the context, tumor type, and the interaction between the MSCs and tumor cells [59]. Currently, studies investigating the tumorigenic potential of MSC-ext remain limited. However, studies on similar cell-free therapies utilizing cellular components, such as extracellular vesicles, have shown no evidence of tumorigenesis or ectopic tissue formation under controlled experimental conditions [60]. Furthermore, MSC-ext can be stored and transported at −20 °C, making it easier to handle as a therapeutic material than MSCs, which are living cells. Owing to these advantages over MSC transplantation, MSC-ext has the potential to become a novel therapeutic strategy for skin wound healing.

## 4. Materials and Methods

### 4.1. Cell Preparation and Culture

The UE7T-13 bone marrow MSC line was obtained from the Riken BioResource Research Center (Tsukuba, Japan). They were established as immortalized cell lines by gene transfection with papillomavirus type 16 protein E7 and human telomerase reverse transcriptase [61]. The UE7T-13 cells were cultured in a α-minimal essential medium (αMEM, Thermo Fisher Scientific, Waltham, MA, USA) containing 15% fetal bovine serum (FBS, Hyclone, Logan, UT, USA), GlutaMax (Thermo Fisher Scientific), and 1% of an antibiotic–antimycotic solution (Nacalai Tesque, Kyoto, Japan). The human dermal fibroblasts (NHDF) (Lonza, Basel, Switzerland) were cultured in a Dulbecco’s modified Eagle’s medium (DMEM, Nacalai Tesque) supplemented with 10% FBS and 1% of an antibiotic–antimycotic solution. The human squamous cell carcinoma cells (HSC-1) were obtained from the JCRB Cell Bank (Osaka, Japan) and cultured in a DMEM containing 20% FBS and 1% of an antibiotic–antimycotic solution. The vascular endothelial cell line, MS1, was a kind gift from Professor Tetsuro Watanabe of the Department of Biochemistry at Tokyo Medical and Dental University. The MS1 was cultured in a αMEM containing 10% FBS and 1% of an antibiotic–antimycotic solution. All cells were maintained in an incubator at 37 °C with 5% CO_2_. When the cells reached 80–90% confluency, they were digested with 0.05% trypsin-ethylenediaminetetraacetic acid (trypsin-EDTA, Nacalai Tesque) and passaged. The medium was changed every 2–3 days. All experiments were performed under sterile conditions.

### 4.2. Preparation of MSC-Ext

The cultured UE7T-13 cells in the logarithmic growth phase (70–80% confluency) were harvested using 0.05% trypsin-EDTA (Nacalai Tesque), washed with a αMEM, and resuspended in PBS at a density of 10⁶ cells/100 μL. The cell suspension subjected to three cycles of freezing and thawing at −80 °C and 37 °C, respectively, to disrupt the cell membranes. After centrifugation at 15,000 rpm for 30 min at 4 °C, the cell debris was removed, and the supernatant was filtered through a 0.22 µm filter (Kurabo, Osaka, Japan) to obtain the cellular extract. The samples were kept at a temperature below 4 °C throughout the procedure. The protein concentration of the MSC-ext was measured using a BCA protein assay kit (Takara Bio, Shiga, Japan).

### 4.3. Water Soluble Tetrazolium (WST)-8 Proliferation Assay

A WST-8 proliferation assay was performed to examine the proliferation of the NHDF, HSC-1, and MS1 cells. The cells were inoculated in 96-well plates (3000 cells/well) with a αMEM containing 1% FBS (100 μL/well) and different concentrations of MSC-ext/PBS (100 μL/well). Figure 2A shows the final concentrations. Cell proliferation was monitored after 72 h using the WST-8 assay reagent (Dojindo, Kumamoto, Japan). The cell proliferation rate was expressed as the absorbance at 450 nm using a microplate reader (SpectraMax M5; Molecular Devices, San Jose, CA, USA).

### 4.4. Transwell Migration Assay

Cell migration was examined using the Boyden chamber method using Chemotaxicell (Kurabo) with a pore size of 8 μm. First, 100 μL/well of cell suspension was inoculated into the upper chamber in 24-well plates (IWAKI, Tokyo, Japan) at a concentration of 1 × 10^5^ cells/mL (final 1 × 10^4^ cells/well), then 400 μL of a serum-free medium containing MSC-ext (350 μg/mL) was added to the lower chamber for the experimental group, while PBS was added to the control group. Specifically, the final concentration of MSC-ext was 280 μg/mL. The cells were incubated at 37 °C with 5% CO_2_ for 18 h. After incubation, the medium in the upper chamber was removed and the membrane was washed twice with PBS. The unmigrated cells were swabbed, and the membranes containing migrated cells were fixed with 4% paraformaldehyde (PFA, Nacalai Tesque) for 20 min and stained with 0.1% crystal violet (Nacalai Tesque) for 30 min. Finally, the cells that migrated through the membrane were counted using a microscope. Each set of experiments was repeated thrice. A quantitative analysis was performed using ImageJ software (1.53K, NIH Image, Bethesda, MD, USA).

### 4.5. Scratch Wound Closure Assay

In the wound healing assay, the cells were cultured in 24-well plates (1 × 10^5^ cells/well). When the cells reached 80–90% confluency, they were scraped and washed with PBS three times, then cultured for 18 h with a medium containing 1% FBS and MSC-ext (with the final concentration of 350 μg/mL). The co-cultures were then visualized using an inverted microscope (CKX53, Olympus, Tokyo, Japan) at 0 and 18 h. The area of the gaps was measured using Image J software.

### 4.6. Skin Defect Model and Experimental Design

The animal experimental protocol was approved by the Animal Experiment Committee of Osaka Dental University (Animal Experiment Approval Nos. 22-02026, 23-04011). Hos:HR mice were used, as the observation and quantification of the size of gross skin defects can easily be performed with a high reproducibility. The Hos:HR mice (female, 8 weeks old, from Shimizu Laboratory Materials Co. Ltd., Tokyo, Japan) were anesthetized via the intraperitoneal administration of metronidazole (Zenoaq, Fukushima, Japan), midazolam (Astellas Pharma Inc., Tokyo, Japan), and butorphanol (Meiji Seika Pharma Co., Ltd., Tokyo, Japan). Skin defects, 4 mm in diameter, were created on the backs of the mice using a Biopsy Punch (Kai Medical, Tokyo, Japan). MSC-ext (1750 μg/mL, 200 μL/defect) or PBS (200 μL/defect) was administered for five consecutive days. The injections were made in the subcutaneous area of the circular wound, and four points of 0 degree, 90 degrees, 180 degrees, and 270 degrees were selected around the circular wound (50 µL/point, 200 µL/wound). The defects were photographed using a digital camera (Canon EOS40D, Canon, Tokyo, Japan) every 1–2 days for 13 days after defect creation. The percentage of skin defect closure was determined relative to the initial defect size measured on day 0. For the 0–13 days observations, we used 6 animals (12 defect sites), while for the tissue section analysis, we used 4 animals (8 defect sites).

### 4.7. Histological Staining

On day 5, the tissue samples around the skin defects were fixed in 4% PFA for 24 h and washed with PBS. The paraffin-embedded sections were prepared by hematoxylin–eosin (H&E) staining, Azan staining, and Picrosirius red staining, which were performed by KAC Co. (Shiga, Japan). The number of inflammatory cells was measured in the tissue sections of the H&E-stained defects. Azan and Sirius red staining were also performed to evaluate the degree of collagen formation. The images were obtained using a polarization microscope (Nikon, Tokyo, Japan).

### 4.8. Immunofluorescence Staining

Ki67 staining was performed to assess the proliferation of the NHDF, HSC-1, and MS1 cells. The cells (3000 cells/well) were seeded into slide chambers (#192008, WASTON, Tokyo, Japan) with a medium containing 1% FBS, supplemented with MSC-ext (with a final concentration of 350 μg/mL) or PBS. The cells were then incubated for 24 h prior to the analysis. The cells were fixed in 4% PFA for 20 min and washed with PBS. After blocking with a 10% goat serum in PBS, a Ki67 antibody conjugated to Alexa Fluor 488 (1:200, #11882; Cell Signaling Technology, Danvers, MA, USA) was applied to the cells. After washing and nuclear staining with DAPI-Fluoromount-G incubation (Southern Biotech, Birmingham, AL, USA), the cells were examined under a BZ-X800 microscope (Keyence, Tokyo, Japan). Ki67 staining was also performed to examine the proliferation of the cells within the tissue samples. The paraffin sections were deparaffinized, rehydrated, and antigen-activated using HistoVT One (Nacalai Tesque). The samples were permeabilized with 0.1% Triton X-100 in 1× PBS and blocked with 5% goat serum in 1x PBS. Subsequently, a Ki67 antibody (1:100) was applied to the sections. After washing and nuclear staining, the sections were examined under a microscope. A quantitative analysis of the Ki67-positive cells was performed using ImageJ software, each experiment utilized three independent cultures, and five random fields per cell type were analyzed for a statistical evaluation. The experiment was repeated three times, resulting in a total of nine independent cell cultures analyzed across the three cell types.

### 4.9. Statistical Analysis

All data are expressed as the mean ± standard deviation (SD). All data were processed with ImageJ software. All statistical analyses were performed using GraphPad Prism version 8 (Boston, MA, USA). The Student’s *t*-test was used for two-group comparisons, and Dunnett’s test was used for a post hoc analysis following a one-way analysis of variance (ANOVA) for comparisons of more than three groups to verify the statistical significance. The values of *p* < 0.05 were considered statistically significant.

## 5. Conclusions

This study showed that the cell extracts from immortalized MSCs promoted the proliferation and/or migration of the fibroblasts, epithelial cells, and endothelial cells involved in skin wound healing and accelerated the closure of skin defects when administered to a skin defect in the mouse model. Thus, MSC-ext serves as a new cell-free treatment for improving skin wound healing. Further optimization of the mechanism of action and delivery of MSC-ext may lead to the development of more efficient and safer skin regeneration therapies.

## Figures and Tables

**Figure 1 ijms-25-13745-f001:**
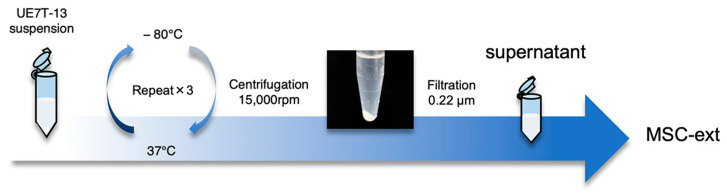
Schematic view of the extraction process for mesenchymal stem cell extract (MSC-ext).

**Figure 2 ijms-25-13745-f002:**
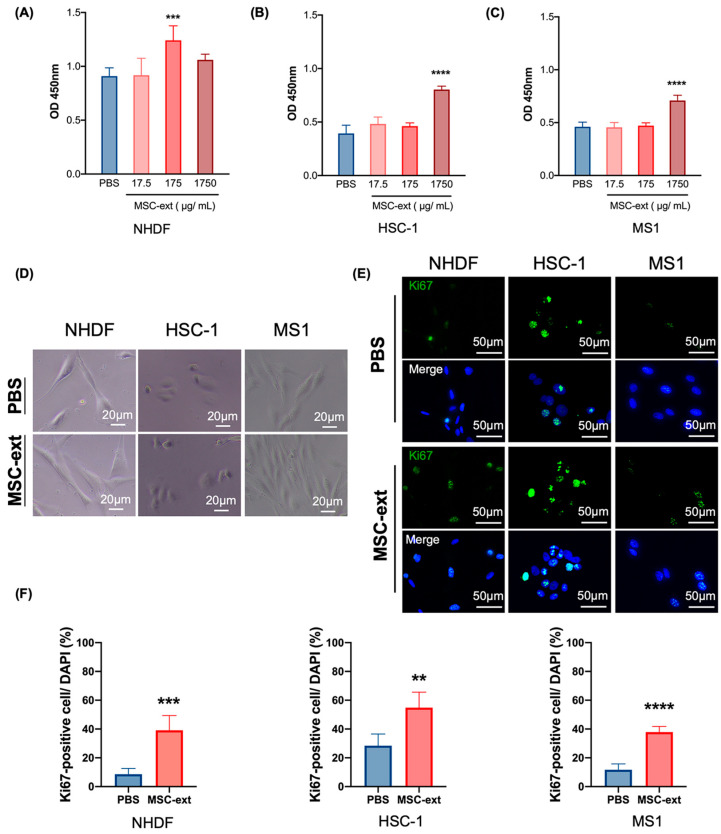
The effect of MSC-ext one cell proliferation. Cell viability was determined by WST-8 at 72 h after MSC-ext addition. Results of viability assay from fibroblasts (NHDF, *n* = 5) (**A**), epithelial cells (HSC-1, *n* = 5) (**B**), and endothelial cells (MS1, *n* = 4) (**C**) are shown. One-way ANOVA followed by Dunnett’s post hoc test (compared with PBS) was used. MSC-ext increased the proliferation of NHDF, HSC-1, and MS1. (**D**) Images of NHDF, HSC-1, and MS1 using phase-contrast microscopy at 24 h after MSC-ext addition are shown. (**E**) Images of Ki67 staining in NHDF, HSC-1, and MS1 at 24 h after MSC-ext addition are shown. Upper and lower panels are from the phosphate-buffered saline (PBS) and MSC-ext groups, respectively. More Ki67-positive cells were found in the MSC-ext groups than in the PBS groups. (**F**) Percentage of the Ki67-positive cells in NHDF, HSC-1, and MS1 are shown (*n* = 5). Student’s *t*-test (comparison with PBS) is used for statistical analysis. For all three cell types, the Ki67-positive cell ratio was higher in the MSC-ext groups than in the PBS groups. All index measurements were taken at five randomly selected sites for statistical analysis. ** *p* < 0.01, *** *p* < 0.001, and **** *p* < 0.0001. All data are expressed as mean ± standard deviation (SD).

**Figure 3 ijms-25-13745-f003:**
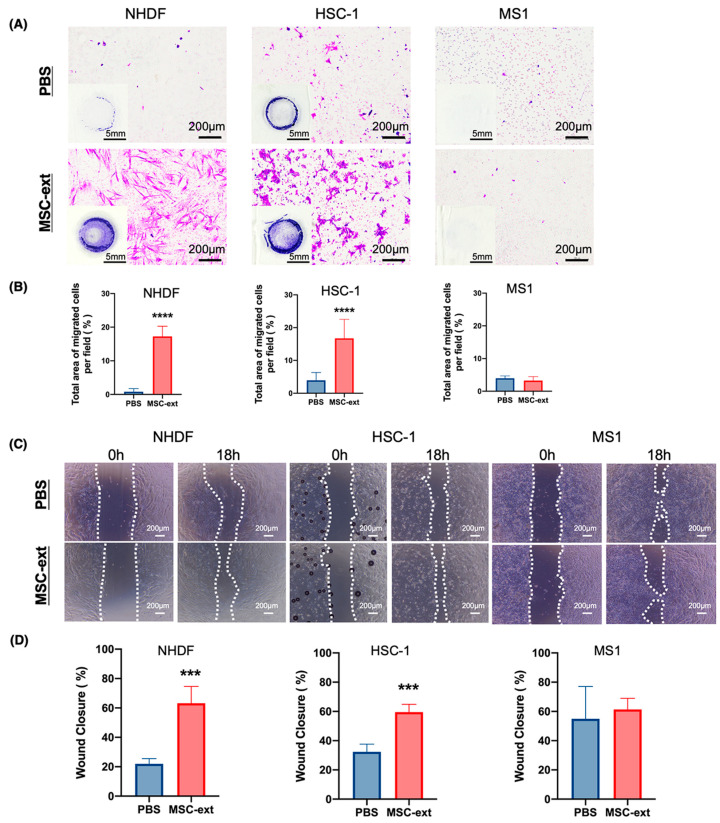
The effect of MSC-ext on cell migration. (**A**) Images of transwell membrane in migration assays from NHDF, HSC-1, and MS1 are shown. The cells that migrated toward MSC-ext or PBS were stained with crystal violet after 18 h. (**B**) The quantification of migrated cells is demonstrated as cells per observed field. All index measurements were taken at ten randomly selected sites under 10× magnification for statistical analysis, *n* = 10. (**C**) Representative phase-contrast microscope images showing the area covered by the cells at 0, and 18 h after wounding under 4× magnification. (**D**) Wound Closure was determined by the rate of cells moving towards the scratched area at a given time, *n* = 4. MSC-ext increased the number of migrated cells in NHDF and HSC-1. *** *p* < 0.001, and **** *p* < 0.0001. Student’s *t*-test (comparison with PBS) is used for statistical analysis. All data are expressed as mean ± SD.

**Figure 4 ijms-25-13745-f004:**
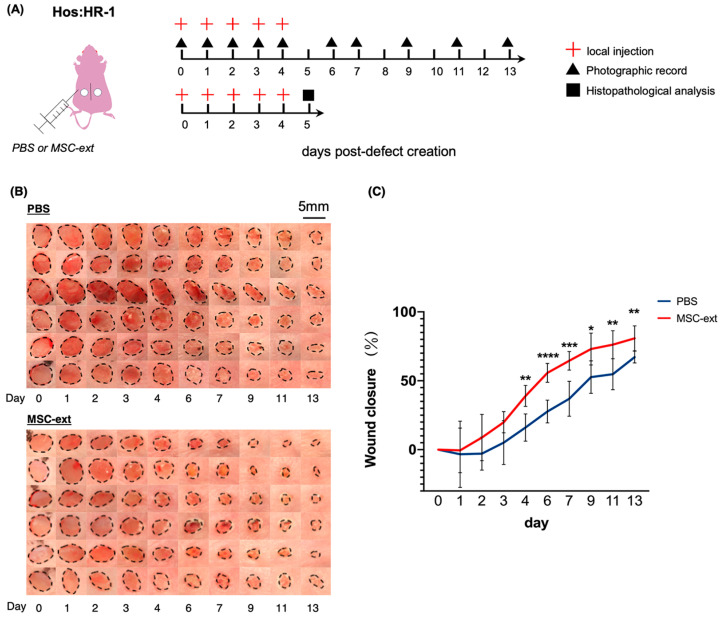
The effect of MSC-ext on mouse skin wound healing. (**A**) Schematic diagram of MSC-ext or PBS injection in mice skin defect model. (**B**) Images of skin defects from PBS (upper) and MSC-ext (lower) are shown. The margins of skin defects were depicted by dotted lines. (**C**) The quantification of the healing of skin defects is shown as the percentage of the defect area on day 0. MSC-ext injection around the skin defect accelerated the healing of the wound compared with PBS. * *p* < 0.05, ** *p* < 0.01, *** *p* < 0.001, and **** *p* < 0.0001. Student’s *t*-test (compared with PBS at each time point) is used for statistical analysis. Defects were prepared on the left and right sides of each mouse. Three mice were used for the PBS and MSC-ext groups, respectively (*n* = 6: 6 defects per group). All data are expressed as mean ± SD.

**Figure 5 ijms-25-13745-f005:**
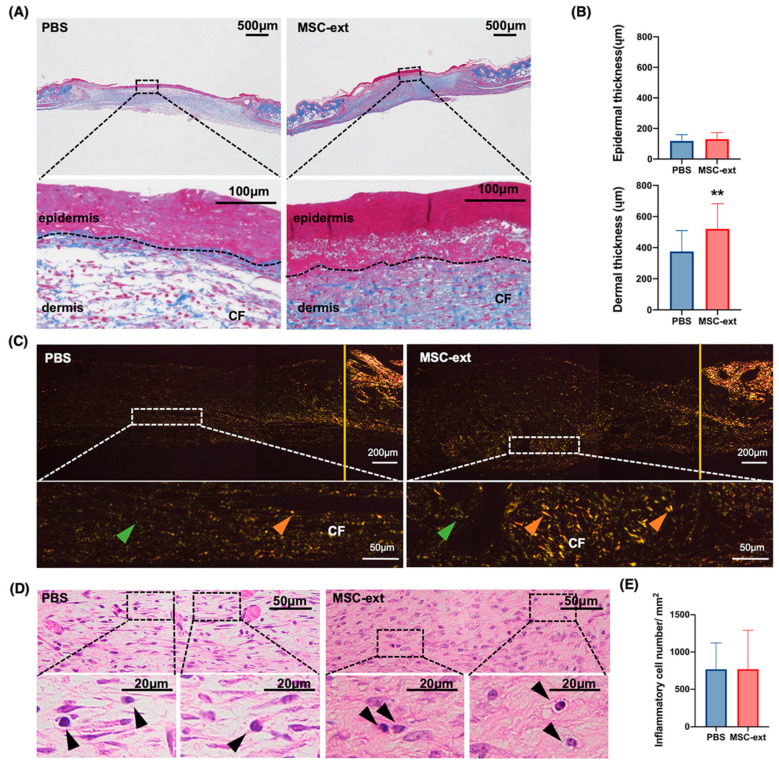
Histologic effects of MSC-ext on wound healing. (**A**) Azan staining images of skin defect at day 5 are shown. CF: collagen fibers. Upper panels are at a lower magnification, and lower panels are magnified images of the black boxes indicated in the upper panels. More areas with blue staining, indicating mature collagen formation, are found in the MSC-ext sections compared to the PBS-treated sections. (**B**) The quantifications of epidermal thickness and dermal thickness are shown. Dermal thickness was higher in the MSC-ext group than in the PBS group. (**C**) Distribution of collagen under polarized light microscopy, with representative areas shown. Yellow solid vertical lines indicate the margin of the skin defect. Upper panels are at a lower magnification, and lower panels are magnified images of the white boxes indicated in the upper panels. Type I collagen—intensely red–orange (indicated with orange arrow heads). Type III collagen—green (indicated with green arrow heads). More polarized mature collagen-like structures were observed in the MSC-ext sections compared to the PBS sections. (**D**) Representative images of hematoxylin–eosin (H&E) stained sections from the PBS group and the MSC-ext group are shown. The boxed areas display higher magnification views of the images. Black arrow heads: inflammatory cells. (**E**) The number of inflammatory cell infiltrates did not show any statistical difference between the PBS and MSC-ext groups. Histological images were taken from five randomly selected sites and analyzed for statistical purposes. ** *p* < 0.01, the Student’s *t*-test (comparison with PBS) is used for statistical analysis. The control and experimental groups were derived from the left and right sides of the skin defects in two mice, respectively (*n* = 4: 4 defects per group). All data are expressed as mean ± SD.

**Figure 6 ijms-25-13745-f006:**
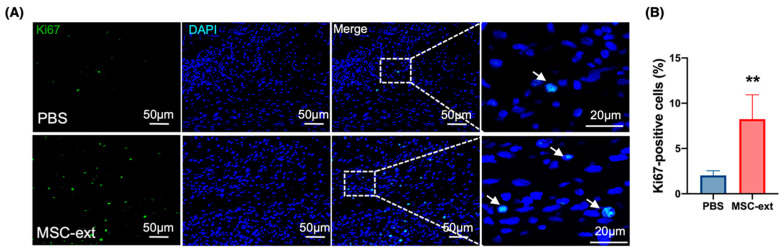
Effect of MSC-ext cell proliferation in wound tissue. (**A**) Cell proliferation in the dermis on day 5 post-injury was determined by Ki67 immunostaining. Upper and lower panels are from the PBS and MSC-ext groups, respectively. As indicated by white arrows, more Ki67-positive cells were found in the dermis treated with MSC-ext than in those treated with PBS. (**B**) The quantification of the Ki67-positive cell ratio in the healing dermis from the PBS and MSC-ext-treated mice is shown. The Ki67-positive cell ratio was higher in the MSC-ext group than in the PBS group. Fluorescence images were taken from three randomly selected sites and analyzed for statistical purposes. ** *p* < 0.01, and the Student’s *t*-test (comparison with PBS) is used for statistical analysis; three random fields of view were selected per section, and the data were then averaged for a statistical analysis (*n* = 4: 4 defects per group). All data are expressed as mean ± SD.

## Data Availability

The data presented in this study are available upon request to the corresponding author.

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
