# Peer review of "Mesenchymal Stem Cell Extract Promotes Skin Wound Healing"

_ijms, 2024, doi:10.3390/ijms252413745_

Round 1

Reviewer 1 Report (Previous Reviewer 2)

Comments and Suggestions for Authors

The authors addressed the concerns

Author Response

We are pleased to note that the reviewer satisfied with our additional experiments and responses. We would like to express our sincere gratitude to the reviewer for dedicating significant time and effort to provide valuable feedback on our manuscript.

Reviewer 2 Report (New Reviewer)

Comments and Suggestions for Authors

The manuscript by Deng et al. addresses the utilization of substances produced by mesenchymal stem cells (MSCs) in wound healing, a topic that is both current and of significant interest to potential readers. While the manuscript is of generally good quality, it contains several shortcomings that need to be addressed before it can be accepted for publication. These include:

  1. Figure 2 and MSC-Ext Yield:
    Figure 2 demonstrates the positive effects of MSC-Ext on the proliferation/viability of different cell types. For fibroblasts, a moderate concentration appears to be most effective, whereas epithelial and endothelial cells respond positively to the highest concentration, which is close to 2 mg/ml. How many MSCs were processed to obtain this amount of extract? In other words, what is the yield of the process shown in Figure 1?

  2. Migration Assay Positive Controls and Concentrations:
    In the migration assays, I noted the absence of positive controls, which could also help explain the low response of endothelial cells to MSC-Ext treatment. Additionally, the migration experiments used a different concentration of MSC-Ext compared to previous experiments. How was this concentration determined?

  3. Discussion on Extract Composition:
    In the discussion, the authors suggest that the differing responses of cell types in the migration assay could be due to varying sensitivities to the composition of the extract, particularly with respect to the growth factors it contains. Has the composition of MSC-Ext been analyzed or characterized? If not, could the authors perform this analysis? Such data would be an important qualitative parameter to ensure the reproducibility of these experiments.

  4. Comparison to Extracts from Non-Immortalized Cells or Other Sources:
    Both the in vitro and in vivo results suggest that the primary effect of MSC-Ext is to promote cell proliferation and migration, which ultimately leads to faster wound closure in vivo. Have the authors tested extracts from other cell sources, including non-immortalized cells? Could the specific phenotype (e.g., immortalization) of MSCs be responsible for the observed effects? How would extracts from non-immortalized MSCs or entirely different cell types perform? This should be addressed in the discussion.

Overall, the manuscript is an interesting contribution to the field. However, to ensure its quality and address the issues mentioned above, I recommend a major revision before considering the manuscript for acceptance.

Author Response

General Response to editor and reviewers:

We would like to express our sincere gratitude to you and the reviewers for dedicating significant time and effort to provide valuable feedback on our manuscript. We are pleased to note that the first reviewer recognized our additional experiments and addressed the reviewer's concerns. We are also grateful to Reviewer 2 for showing interest in our research and raising insightful questions. Additionally, Reviewer 3 thoroughly examined the content and quality of our manuscript, offering constructive suggestions that significantly improved the quality of our paper. All three reviewers helped us identify areas in the manuscript that required further clarification, emphasized important points, and guided us in improving the overall quality. We deeply appreciate the thorough analysis and feedback provided by all the reviewers. Below is a detailed response to each comment, along with the corresponding revisions. The changes in the manuscript are highlighted in yellow.

Comment #1-1. Figure 2 and MSC-ext Yield: Figure 2 demonstrates the positive effects of MSC-Ext on the proliferation/viability of different cell types. For fibroblasts, a moderate concentration appears to be most effective, whereas epithelial and endothelial cells respond positively to the highest concentration, which is close to 2 mg/ml.

Comment #1-2. How many MSCs were processed to obtain this amount of extract? In other words, what is the yield of the process shown in Figure 1?

Response #1-1. Thank you for your insightful observation. In the present study, 175 ug/mL of MSC-extract promoted the proliferation of dermal fibroblasts, while we have previously reported that the MSC-ext ranging from 100 to 400 µg/mL promoted periodontal ligament fibroblasts [1]. As you mentioned, the effective concentration of MSC-ext for promoting proliferation varies across different cell types, which could be attributed to the complex interactions of the components it contains. Regarding this point, we have also emphasized it in the discussion. (Lines 260-269)

[1] (Original position in References: 32) Peng, Y.; Mesenchymal stem cell-derived protein extract induces periodontal regeneration. Cytotherapy. 2024, doi: 10.1016/j.jcyt.2024.10.003

Response #1-2. We apologize for the lack of detailed description of MSC-ext. Thank you for bringing attention to the critical point about the yield of MSC-ext. For the single extraction procedure, we prepared eight 100 mm culture dishes, seeding with cells at a density of 1×10⁶ cells per dish. After 24 to 48 hours of incubation, we confirmed that the cells had reached 90% confluence. The cells were then harvested and resuspended at a final density of 1×10⁶ cells per 100 µL in PBS, which was used to prepare for the extraction. We collected average 188 ±15.6 µg­ MSC-ext from 106 UE7T-13 cells. The yield may vary based on culture conditions and extraction protocol refinements. If additional details are required, we would be glad to provide a more detailed breakdown. The information about the yield has been added. (Lines 76-77)

Comment #2-1. Migration Assay Positive Controls and Concentrations:
In the migration assays, I noted the absence of positive controls, which could also help explain the low response of endothelial cells to MSC-Ext treatment.

Comment #2-2. Additionally, the migration experiments used a different concentration of MSC-Ext compared to previous experiments. How was this concentration determined?

Response #2-1. We apologize for the lacking the positive control, such as VEGF. We totally agree that the addition of positive control must deepen our insight. Thank you for your valuable comment. During the preliminary stages of both the transwell and wound healing assays, we tested various concentrations of MSC-ext, but none showed a significant effect on promoting endothelial cell migration as shown in Fig3.

We agree that a positive control would help provide a clearer comparison and potentially explain the low response of endothelial cells to MSC-Ext treatment. We will ensure that appropriate positive controls are included in future experiments to address this limitation. This is a limitation of the current study, which we have addressed and discussed in the revised manuscript. (Lines 263-265)

Response #2-2. As for the concentration determined in migration experiment, based on six preliminary MSC-ext extractions, the average yield of MSC-ext was quantified using BCA analysis, with an average of 1880.7 µg/mL per extraction. Considering the minimum yield of approximately 1750 µg/mL, we set 1750 µg/mL as the highest concentration group and used lower concentration gro. After evaluation, we selected a concentration of 350 µg/mL (after 5-fold dilution) for the further experiments.

Comment #3. Discussion on Extract Composition: In the discussion, the authors suggest that the differing responses of cell types in the migration assay could be due to varying sensitivities to the composition of the extract, particularly with respect to the growth factors it contains. Has the composition of MSC-Ext been analyzed or characterized? If not, could the authors perform this analysis? Such data would be an important qualitative parameter to ensure the reproducibility of these experiments.

Response #3. Thank you for your valuable comment. In our previous paper, we have already reported on the content of the proteins and growth factors in the MSC-extract using liquid chromatograph-mass spectrometry (LC-MS/MS) and dot blot-based growth factor protein array analysis [1]. The composition of the top 21 proteins and growth factor of MSC-Ext are shown at below. There are many unknown components in the extract, and under the influence of these complex mechanisms, different cell types may exhibit distinct effects. We could not conclude the dominant protein to show the function in the present study, which is a limitation of this study. However, we fully agree with your suggestion. To highlight this point, we newly added the sentences in discussion part. (Lines 241-250, 332-335)

[1] (Original position in References: 32) Peng, Y.; Mesenchymal stem cell-derived protein extract induces periodontal regeneration. Cytotherapy. 2024, doi: 10.1016/j.jcyt.2024.10.003

a

b

The top 21 proteins detected in the MSC-extract (a) and the composition of growth factors (b) are shown.

Comment #4. Comparison to Extracts from Non-Immortalized Cells or Other Sources: Both the in vitro and in vivo results suggest that the primary effect of MSC-Ext is to promote cell proliferation and migration, which ultimately leads to faster wound closure in vivo. Have the authors tested extracts from other cell sources, including non-immortalized cells? Could the specific phenotype (e.g., immortalization) of MSCs be responsible for the observed effects? How would extracts from non-immortalized MSCs or entirely different cell types perform? This should be addressed in the discussion.

Response #4. We appreciate the important points raised by the reviewer. We acknowledge that the results of this study may have been influenced by the use of immortalized MSCs. Although this study did not validate the effects using primary cultured MSCs, we could not demonstrate any differences in efficacy between immortalized and non-immortalized MSCs. We have added content regarding this section in the discussion. Furthermore, we have added additional references to support these points. [45-49] (Lines 304-317, 324-332)

(Original position in References:

  1. Song, J. Y.; Kang, H. J.; Hong, J. S.; Kim, C. J.; Shim, J. Y.; Lee, C. W.; Choi, J. Umbilical cord-derived mesenchymal stem cell extracts reduce colitis in mice by re-polarizing intestinal macrophages. Sci. Rep. 2017, 7, 9412, doi:10. 1038/s41598-017-09827-5.
  2. Kamprom, W.; Tangporncharoen, R.; Vongthaiwan, N. et al. Enhanced potent immunosuppression of intracellular adipose tissue-derived stem cell extract by priming with three-dimensional spheroid formation. Sci. Rep. 2024, 14, 9084.
  3. Abughanam, G.; Elkashty, O. A.; Liu, Y.; Bakkar, M. O.; Tran, S. D. Mesenchymal Stem Cells Extract (MSCsE)-Based Therapy Alleviates Xerostomia and Keratoconjunctivitis Sicca in Sjogren's Syndrome-Like Disease. Int. J. Mol. Sci. 2019, 20, 4750, doi:10.3390/ijms20194750.
  4. Paprocka, M., Kraskiewicz, H., Bielawska-Pohl, A., Krawczenko, A., Masłowski, L., Czyżewska-Buczyńska, A., Witkiewicz, W., Dus, D., Czarnecka, A. From Primary MSC Culture of Adipose Tissue to Immortalized Cell Line Producing Cytokines for Potential Use in Regenerative Medicine Therapy or Immunotherapy. Int. J. Mol. Sci. 2021;22, 11439. doi:10.3390/ijms222111439
  5. Pan, C., Kumar, C., Bohl, S., Klingmueller, U.,Mann, M. Comparative proteomic phenotyping of cell lines and primary cells to assess preservation of cell type-specific functions. Mol. Cell Proteomics. 2009, 8, 443-450, doi:10.1074/mcp.M800258-MCP200

We appreciate your suggestion and have incorporated a detailed discussion of this limitation and the need for future comparative studies into the revised manuscript.

Additional revisions:

We have some mistakes in the previous manuscript: (1) We have confirmed that "UE7T-13" is the more standard expression (“UE7T13” was used before the change), and we have updated the manuscript to reflect this change. We sincerely apologize for this error. (2) Following the other reviewer’s suggestion, we have also adjusted the layout of Figure 2E and Figure 3A, B. In addition, we have unified the expression and changed “fig.” to “figure”.

Reviewer 3 Report (New Reviewer)

Comments and Suggestions for Authors

  1. What do the yellow text highlights indicate?
  2. There are some minor typing errors and missing spaces: Examples: "day0", "Fig.5D".
  3. Keywords: I suggest adding "proliferation" and "therapeutic potential."
  4. Immortalized MSCs do not age in the same way as natural MSCs, as telomere shortening is avoided and mechanisms preventing senescence are activated. However, this does not mean they are entirely resistant to cellular stress, DNA damage, or epigenetic changes. Immortalization is a useful research tool, but caution should be exercised when using these cells in clinical therapies due to the risk of tumorigenic transformation. Please address this in the discussion, including potential negative implications of using such a model.
  5. Repeated freezing and thawing cycles lead to protein degradation.
    Where did the Authors adopt this protocol from? Do the Authors have any unpublished data identifying the protein components of the extract? Please perform additional analyses, e.g., ELISA or Western blot, to confirm the functionality of at least the proteins discussed, such as FGF, HGF, and VEGF.
  6. Fig.1: Include the expanded form of "UE7T" in the figure legend and provide a detailed description of the procedure under the figure or in the "Materials and Methods" section.
  7. Fig. 2D-E: Add a scale bar to each image. This applies to all other figures as well.
  8. Fig. 2E: Specify how Ki67+ cells were counted. How many fields of view were analyzed? How many independent cell cultures were used?
  9. Fig. 2D-E: How many cells were seeded? The confluence in the images is very low (only a few cells in the field of view). If the same number of cells was seeded and fixed after 24 hours, what caused this difference? Please repeat the staining or provide images with similar confluence and redo the quantification.
  10. Please align the spacing between images across all figures.
  11. "Transwell": Write in lowercase (it is not a trademarked term).
  12. Picrosirius red-stained: This method is used to visualize collagen, particularly: Type I collagen – intensely red-orange. Type III collagen – green (under a polarizing microscope). Please clarify this in the figure legend and indicate these observations on the images.
  13. Lines 108-109: "In the wound healing assay, we show that MSC-ext addition causes cells to migrate faster to close the gap of a scratch in the cell monolayer than PBS-treated cells." For the MS1 line, the result is even lower than in PBS. This sentence needs to be rephrased, clarified, or removed.
  14. Fig.2A-C: Please rearrange the layout. I suggest organizing the results in columns for NHDF, HSC-1, and MS1, with corresponding images and graphs in rows. This will correlate better with Fig.2D, improving clarity for readers. Consider including the images from Fig.2A as insets in the images from Fig.2B.
  15. Fig.4A: Please improve the aesthetics of the diagram. Use the entire width of the figure. Replace the arrows with more visually appealing ones, improve labels, and label the mouse drawings. The diagram should be more visually refined.
  1. Fig. 5A: Please indicate the epidermal and dermal layers in the image. Were the measurements taken from different sections of the same mouse or from multiple mice, or just a single section/sample? This needs to be clarified.
  2. Fig. 5C: What do the thin vertical lines in the image represent? Is it "White dotted lines indicate the margin of the skin defect?" Please specify that these are vertical lines (as there are other similar lines in the image).
  3. Fig. 5D: The font in the image is barely visible; consider changing it to black for better contrast. On what basis do the authors conclude these are collagen fibers? The structures indicated by the arrows resemble clusters of pigment cells or fibrosis (pathological changes). Please consult a pathologist if possible or perform additional analyses. One of the magnified areas in the PBS image comes from a region different from the one marked. Please correctly identify collagen fibers as described in the text. The figure legend suggests the arrows point to inflammatory cells. Is this accurate? The image is ambiguous. Similarly, for fibroblasts, the indicated structures appear to be longitudinally cut blood vessels. Please include higher magnifications, as the arrows are too small and barely visible. These issues need to be verified, clarified, or corrected.
  1. Fig. 5E: How were the inflammatory cells identified? Please include an image illustrating these cells.
  2. Fig. 6A: Remove the word "magnification" from the image and include an appropriate scale bar instead. Why is there a lower cell count in the PBS image? What caused this difference? The result is unreliable, as a higher cell density increases the likelihood of finding Ki67+ cells. Please repeat the experiment/staining or reanalyze the existing data, showing images with similar confluence, and perform proper quantifications. From how many cells was the percentage estimated? How many cells were counted?
  1. Fig. 6B: How were the cells counted? Does "three randomly selected sites" mean three sections from the same mouse or from different mice? Does N=4 represent different animals or different sections?
  2. Discussion: Please relate and discuss your results on migration/proliferation in the context of other publications on this topic. Look for literature discussing the effects of MSCs on these processes. MSC extracts may contain not only proteins but also components such as miRNA, which undoubtedly has an impact. Highlight the anti-inflammatory properties of MSCs and their chemoattractant effects on host/mouse cells. The current discussion is highly speculative and does not provide a comprehensive picture of MSC activity. A more thorough analysis of the available literature is needed, along with confirmation of the presence of at least a few key proteins as indicated in point 5 of this review.
  1. Line 204: It repeats the content of line 198. Please rephrase or remove it.
  1. Line 224: You could consider using a method such as Luminex-multiplex, which allows for the detection of dozens of proteins in a single sample.
  2. Lines 266-280: Please explain why the authors chose this model and why the experiments were conducted for only 13 days. Why were young mice used?
  3. Line 297: MSCs are not tumorigenic (unlike iPS cells). Please cite an appropriate source to clarify and support this statement.
  4. Materials and Methods, Line 331: “(100 μL/well) and different concentrations of MSC-ext/PBS (100 μL/well)." Were the concentrations properly prepared? Adding 100 μL of extract to 100 μL of culture medium dilutes the protein concentration by half (!). Are the concentrations reported in the “Results” section accurate? What was the initial concentration of MSC-ext?
  5. Line 338: "per well at a concentration of 1 × 10⁵ cells/mL” – Media were added to the well, which would have altered the final cell concentration (number of cells/mL). Please clarify or provide the total number of cells per well. “(350 μg/mL concentration)” – Please specify the final concentration, taking into account the total volume of fluid in the well. Line 340: “200 μL of MSC-ext (350 μg/mL concentration)” – Was this the final concentration? Please verify concentration issues throughout the manuscript (not all instances are highlighted in this review), including lines 386-387 and others.
  1. Line 372: "In some experiments, the mice were euthanized on postoperative day 5." How many experiments were conducted in total? In Figure 4B, all time points (days 0-13) are shown for all 6 mice in one experiment. This is inconsistent with the text. Please revise or clarify this discrepancy.

Author Response

General Response to editor and reviewers:

We would like to express our sincere gratitude to you and the reviewers for dedicating significant time and effort to provide valuable feedback on our manuscript. We are pleased to note that the first reviewer recognized our additional experiments and addressed the reviewer's concerns. We are also grateful to Reviewer 2 for showing interest in our research and raising insightful questions. Additionally, Reviewer 3 thoroughly examined the content and quality of our manuscript, offering constructive suggestions that significantly improved the quality of our paper. All three reviewers helped us identify areas in the manuscript that required further clarification, emphasized important points, and guided us in improving the overall quality. We deeply appreciate the thorough analysis and feedback provided by all the reviewers. Below is a detailed response to each comment, along with the corresponding revisions. The changes in the manuscript are highlighted in yellow.

Comment #1. What do the yellow text highlights indicate?

Response: We sincerely apologize for this oversight. After submission, we promptly contacted the editor regarding this issue. It appears that a system error may have caused the reviewer to access an incomplete version of the manuscript. The sections highlighted in yellow now indicate the revised portions.

Comment #2. There are some minor typing errors and missing spaces: Examples: "day0", "Fig.5D".

Response: We sincerely apologize for our oversight. We greatly appreciate your careful observation. The necessary corrections have been made. (Line173)

Comment #3. Keywords: I suggest adding "proliferation" and "therapeutic potential."

Response: Thank you very much for your valuable suggestion. We have accepted your feedback and added "proliferation" and "therapeutic potential to the keywords accordingly.

Comment #4. Immortalized MSCs do not age in the same way as natural MSCs, as telomere shortening is avoided and mechanisms preventing senescence are activated. However, this does not mean they are entirely resistant to cellular stress, DNA damage, or epigenetic changes. Immortalization is a useful research tool, but caution should be exercised when using these cells in clinical therapies due to the risk of tumorigenic transformation. Please address this in the discussion, including potential negative implications of using such a model.

Response: Thank you very much for your valuable feedback. It has helped make the wording of this study more precise. We have incorporated additional content into the discussion section to address this point. (Lines 305-317, 328-332) Furthermore, we have added additional references to support these points [45-50].

References:

  1. Song, J. Y.; Kang, H. J.; Hong, J. S.; Kim, C. J.; Shim, J. Y.; Lee, C. W.; Choi, J. Umbilical cord-derived mesenchymal stem cell extracts reduce colitis in mice by re-polarizing intestinal macrophages. Sci. Rep. 2017, 7, 9412, doi:10. 1038/s41598-017-09827-5.
  2. Kamprom, W.; Tangporncharoen, R.; Vongthaiwan, N. et al. Enhanced potent immunosuppression of intracellular adipose tissue-derived stem cell extract by priming with three-dimensional spheroid formation. Sci. Rep. 2024, 14, 9084.
  3. Abughanam, G.; Elkashty, O. A.; Liu, Y.; Bakkar, M. O.; Tran, S. D. Mesenchymal Stem Cells Extract (MSCsE)-Based Therapy Alleviates Xerostomia and Keratoconjunctivitis Sicca in Sjogren's Syndrome-Like Disease. Int. J. Mol. Sci. 2019, 20, 4750, doi:10.3390/ijms20194750.
  4. Paprocka, M., Kraskiewicz, H., Bielawska-Pohl, A., Krawczenko, A., Masłowski, L., Czyżewska-Buczyńska, A., Witkiewicz, W., Dus, D., Czarnecka, A. From Primary MSC Culture of Adipose Tissue to Immortalized Cell Line Producing Cytokines for Potential Use in Regenerative Medicine Therapy or Immunotherapy. Int. J. Mol. Sci. 2021;22, 11439. doi:10.3390/ijms222111439
  5. Pan, C., Kumar, C., Bohl, S., Klingmueller, U.,Mann, M. Comparative proteomic phenotyping of cell lines and primary cells to assess preservation of cell type-specific functions. Mol. Cell Proteomics. 2009, 8, 443-450, doi:10.1074/mcp.M800258-MCP200
  6. Shitova, M.; Alpeeva, E.; Vorotelyak, E. Review of hTERT-Immortalized Cells: How to Assess Immortality and Confirm Identity. Int. J. Mol. Sci. 2024, 25, 13054, doi:10.3390/ijms252313054

Comment #5. Repeated freezing and thawing cycles lead to protein degradation. Where did the Authors adopt this protocol from? Do the Authors have any unpublished data identifying the protein components of the extract? Please perform additional analyses, e.g., ELISA or Western blot, to confirm the functionality of at least the proteins discussed, such as FGF, HGF, and VEGF.

Response: Thank you for your insightful comment. We have previously published two studies on MSC-ext. In previous study, protein array analysis, ELISA and LC-MS/MS were employed to comprehensively analyze the components of MSC-ext, including growth factors and proteins [1, 2]. The findings revealing that the major growth factors include bFGF and HGF, while VEGF is present in relatively smaller amounts. These results align with our understanding of the composition of MSC-ext, providing valuable context for its biological effects. We have added this reference and further clarified these points in the discussion to address your concern. (Lines 245-250)

a

b

The composition of growth factors (a) and the top 21 proteins detected in the MSC-extract (b) are shown.

References:

31.Peng, Y.; Iwasaki, K.; Taguchi, Y.; Umeda, M. The Extracts of Mesenchymal Stem Cells Induce the Proliferation of Periodontal Ligament Cells. J. Osaka Dent. Univ. 2023, 57, 119–124, doi:10.18905/jodu.57.1_119.

32.Peng, Y.; Iwasaki, K.; Taguchi, Y.; Ishikawa, I.; Umeda, M. Mesenchymal stem cell-derived protein extract induces periodon-tal regeneration. Cytotherapy. 2024, in press, doi: 10.1016/j.jcyt.2024.10.003

Comment #6. Fig.1: Include the expanded form of "UE7T" in the figure legend and provide a detailed description of the procedure under the figure or in the "Materials and Methods" section.

Response: Thank you for your valuable feedback. We sincerely apologize for this error. Upon further check, we have confirmed that "UE7T-13" is the more standard expression, and we have updated the manuscript to reflect this change throughout. Additionally, we have provided a more detailed description of the extraction process, which has now been added to the Materials and Methods section. (Line383-389)

Comment #7. Fig. 2D-E: Add a scale bar to each image. This applies to all other figures as well.

Response: Thank you for your valuable suggestion. We have now added the scale bar to the figure as requested.

Comment #8. Fig. 2E: Specify how Ki67+ cells were counted. How many fields of view were analyzed? How many independent cell cultures were used?

Response: Thank you for your insightful feedback. Cell counts were performed using ImageJ as follows: the total number of nuclei stained with DAPI was used as the denominator, while Ki67-positive cells overlapping with DAPI staining were counted as the numerator to calculate the percentage. Each experiment utilized three independent cultures, and five random fields per cell type were analyzed for statistical evaluation. The experiment was repeated three times, resulting in a total of nine independent cell cultures analyzed across three cell types. We have clarified these details in the revised manuscript for transparency. (Lines 463-466)

Comment #9. Fig. 2D-E: How many cells were seeded? The confluence in the images is very low (only a few cells in the field of view). If the same number of cells was seeded and fixed after 24 hours, what caused this difference? Please repeat the staining or provide images with similar confluence and redo the quantification.

Response: We sincerely apologize for any confusion caused and deeply appreciate your constructive feedback.

Regarding Figure 2D, we apologize for the poor quality of the images and have chosen a better representative area and improved the image clarity. The cells were initially seeded in a slide chamber with a base area of 70 mm² at a density of 3000 cells per chamber. Based on the different cell types and sizes, the density performance will change after 24 hours of MSC-ext stimulation.

For Figure 2E, the combination of higher magnification and representative image selection may have led to misunderstanding. To resolve this, we have replaced the NHDF representative image to ensure consistent confluency across the three cell types and have recalculated the related statistical analysis for NHDF. Additionally, we have included a lower magnification overview image for your reference.

(Modifications are reflected in Figure 2D, Figure 2E for NHDF cells and Figure 2F for statistical analysis of NHDF cells.)

Comment #10. Please align the spacing between images across all figures.

Response: We sincerely appreciate your valuable feedback. We have aligned all the space between images across all figures.

Comment #11. "Transwell": Write in lowercase (it is not a trademarked term).

Response: Thank you for your suggestions. We have made the necessary revisions.

Comment #12. Picrosirius red-stained: This method is used to visualize collagen, particularly: Type I collagen – intensely red- orange. Type III collagen – green (under a polarizing microscope). Please clarify this in the figure legend and indicate these observations on the images.

Response: We sincerely appreciate your detailed suggestions. The requested addition has been included in the legend of Figure 5. (Line 205-206)

Comment #13. Lines 108-109: "In the wound healing assay, we show that MSC-ext addition causes cells to migrate faster to close the gap of a scratch in the cell monolayer than PBS-treated cells." For the MS1 line, the result is even lower than in PBS. This sentence needs to be rephrased, clarified, or removed.

Response: Thank you for your careful observations. They were of great help to us. We have made some modifications to the content. (Line139-142)

Comment #14. Fig.2A-C: Please rearrange the layout. I suggest organizing the results in columns for NHDF, HSC-1, and MS1, with corresponding images and graphs in rows. This will correlate better with Fig.2D, improving clarity for readers. Consider including the images from Fig.2A as insets in the images from Fig.2B.

Response: Thank you very much for your suggestion. We have adopted part of it by organizing the results of NHDF, HSC-1, and MS1 with corresponding images and graphs in rows. However, regarding the suggestion to include the images from Figure 2A as illustrations within the images of Figure 2B, we remain reserved. This is because the CCK-8 results primarily demonstrate the effects of different concentrations on cell viability (Figures 2A-C), whereas the results from cell staining and microscopic observations focus on morphological changes in the cells (Figures 2D, E). We believe that presenting them separately is more appropriate.

However, we were inspired to change the layout of figures 3A and 3B. We merged then to make it clearer for readers.

Nonetheless, we sincerely appreciate your suggestion, which has helped us improve the quality of our figure presentation.

Comment #15. Fig.4A: Please improve the aesthetics of the diagram. Use the entire width of the figure. Replace the arrows with more visually appealing ones, improve labels, and label the mouse drawings. The diagram should be more visually refined.

Response: We are grateful for your valuable suggestions and feedback. We have made the corresponding modifications to Figure 4A accordingly.

Comment #16. Fig. 5A: Please indicate the epidermal and dermal layers in the image. Were the measurements taken from different sections of the same mouse or from multiple mice, or just a single section/sample? This needs to be clarified.

Response: We sincerely appreciate your suggestion. Annotations have been added to Figure 5A, and the measurements were derived from multiple mice. We have added annotations to the legends of both Figures 4 and 5. (Lines 176-177, 212-214)

Comment #17. Fig. 5C: What do the thin vertical lines in the image represent? Is it "White dotted lines indicate the margin of the skin defect?" Please specify that these are vertical lines (as there are other similar lines in the image).

Response: Thank you for your reminder. Thank you for your comment. As you mentioned, the thin vertical lines in the image represent the margins of the skin defect. To avoid any potential misunderstanding, we have replaced the white dotted lines with yellow solid vertical lines, clarifying this in the figure legend. (Figure 5C).

Comment #18. Fig. 5D: The font in the image is barely visible; consider changing it to black for better contrast. On what basis do the authors conclude these are collagen fibers? The structures indicated by the arrows resemble clusters of pigment cells or fibrosis (pathological changes). Please consult a pathologist if possible or perform additional analyses. One of the magnified areas in the PBS image comes from a region different from the one marked. Please correctly identify collagen fibers as described in the text. The figure legend suggests the arrows point to inflammatory cells. Is this accurate? The image is ambiguous. Similarly, for fibroblasts, the indicated structures appear to be longitudinally cut blood vessels. Please include higher magnifications, as the arrows are too small and barely visible. These issues need to be verified, clarified, or corrected.

Response: Thank you for your comment. We have revised the font as suggested. Regarding the identification of collagen fibers, we consulted professor Tominaga of the Department of Pathology at Osaka Dental University, who pointed out that, as you have concerns, it is not possible to confirm the presence of collagen fibers based solely on H&E staining. The professor also suggested that the cells in the magnified images are inflammatory cells. However, to better identify specific types of inflammatory cells, it is necessary to use immunohistochemistry or other methods for staining and identification. Due to a lack of sufficient evidence and to prevent potential controversy among readers, we have decided not to discuss the identification of collagen fibers and specific types of inflammatory cells in the H&E-stained sections. (Lines 192-195)

Comment #19. Fig. 5E: How were the inflammatory cells identified? Please include an image illustrating these cells.

Response: We sincerely appreciate your valuable feedback. As mentioned in the previous response, the cell types should be assessed in combination with other staining methods. Therefore, in present study, we do not focus on identifying specific types of inflammatory cells. In the actual process of inflammatory cell counting, we identified and assessed the cells according to the following criteria: neutrophils (multinucleated, lobulated nuclei, 3–5 lobes, with light pink cytoplasm), lymphocytes (single nucleus, round and deeply stained, with minimal cytoplasm), plasma cells (nucleus located peripherally, strongly basophilic cytoplasm), eosinophils (distinct bilobed nuclei, with cytoplasm containing prominent red coarse granules) and macrophages (cells are vary in size and have irregularly shaped, oval or reniform nuclei). Since the magnified tissue sections do not show all representative inflammatory cells, we have included additional representative inflammatory cells in the images below for your reference.

Comment #20. Fig. 6A: Remove the word "magnification" from the image and include an appropriate scale bar instead. Why is there a lower cell count in the PBS image? What caused this difference? The result is unreliable, as a higher cell density increases the likelihood of finding Ki67+ cells. Please repeat the experiment/staining or reanalyze the existing data, showing images with similar confluence, and perform proper quantifications. From how many cells was the percentage estimated? How many cells were counted?

Response: Thank you very much for your observation. We followed your advice, removed the magnification, and added a scale bar. In the tissue sections, we found that the number of cells per unit area in the PBS group was lower than in the MSC-ext group (approximately 100 fewer cells per unit area in the PBS group compared to the MSC-ext group). We speculate that this result is due to the cell migration and proliferation-promoting effects of MSC-ext. To address your concerns, we have provided Ki67-stained images from four tissue samples for your reference. Additionally, following your suggestion, we have selected images with similar levels of confluence for Figure 6.

Since it is not feasible to standardize the cell number in tissue samples, we believe calculating the Ki67 positivity rate is currently the most suitable statistical approach. Our statistical analysis was performed based on four tissue sections per group, with three random fields of view selected per section (magnified at 40x objective, with an average of 300–400 cells per field). The data were then averaged for statistical analysis. (Line228-229)

Comment #21. Fig. 6B: How were the cells counted? Does "three randomly selected sites" mean three sections from the same mouse or from different mice? Does N=4 represent different animals or different sections?

Response: Thank you very much for your question. Regarding cell counting, we used the ImageJ analysis software. DAPI-stained nuclei were used as the denominator, with Ki67-positive nuclei overlapping with DAPI serving as the numerator to calculate the percentage. We analyzed a total of four skin tissue sections from the left and right sides of two animals per group. For each section, three random fields of view were selected, and the averages were used for analysis. (Line228-229)

Comment #22. Discussion: Please relate and discuss your results on migration/proliferation in the context of other publications on this topic. Look for literature discussing the effects of MSCs on these processes. MSC extracts may contain not only proteins but also components such as miRNA, which undoubtedly has an impact. Highlight the anti-inflammatory properties of MSCs and their chemoattractant effects on host/mouse cells. The current discussion is highly speculative and does not provide a comprehensive picture of MSC activity. A more thorough analysis of the available literature is needed, along with confirmation of the presence of at least a few key proteins as indicated in point 5 of this review.

Response: Thank you for your valuable feedback. We have revised the Discussion section to provide a more comprehensive explanation of the effects of mesenchymal stem cells (MSCs) on cell migration and proliferation, with a detailed discussion based on existing literature. The revised discussion includes the following improvements:

1)Comparison with Existing Studies.

The Discussion now includes a comparison of how MSCs and their extracts influence cell migration and proliferation, highlighting similarities and differences between our findings and those of previous studies. (Line 275-294)

2)Role of Non-Protein Components and Anti-Inflammatory and Chemotactic Properties.

Additional content discussing the anti-inflammatory and chemotactic properties of MSC extracts has been included. (Line 343-345)

3)Validation of Key Proteins.

In response to the reviewer’s comment on point 5, we have confirmed the presence of proteins contains in the MSC extract through methods described in the Materials and Methods section and further analyzed their potential mechanisms of action in the Discussion. For future studies, we plan to perform neutralizing antibody experiments to identify the specific components responsible for the main effects of the extract. (Line 245-250)

This has also been addressed in the revised discussion. We appreciate your suggestions, which have significantly enhanced the comprehensiveness and rigor of our discussion.

Comment #23. Line 204: It repeats the content of line 198. Please rephrase or remove it.

Response: Thank you for your insightful feedback. We have revised the relevant expressions accordingly. (Line 238-240)

Comment #24. Line 224: You could consider using a method such as Luminex-multiplex, which allows for the detection of dozens of proteins in a single sample.

Response: Thank you very much for your professional suggestion. As addressed in the response to Comment 5, we previously employed LC-MS/MS and protein array in our earlier study to detect 4,388 proteins, which has been documented in the discussion section. (Line 245-250)

Comment #25. Lines 266-280: Please explain why the authors chose this model and why the experiments were conducted for only 13 days. Why were young mice used?

Response: Thank you for your question.

Reason for this model

Hos-HR mice were chosen for their hairless phenotype, which enables direct observation of wound healing without interference from hair regrowth. We initially began with C57BL/6 mice but switched due to the difficulty of observing wounds through regrowing hair. While hair follicle stem cells are important for wound healing, this study focused on evaluating MSC-ext effects in a controlled environment. The hairless nature of Hos-HR mice allowed precise assessment of wound closure and tissue regeneration, making them ideal for accurate and reproducible results.

Reason for 14-day experiment

We conducted a 14-day experiment (days 0-13) because significant changes in skin healing occur in the first 1-2 weeks, including wound repair and cell proliferation. By day 13, wounds were nearly healed, effectively capturing key dynamics of the healing process. Additionally, a 2-week duration balances feasibility, cost, and the ability to evaluate early MSC-ext effects. Long-term effects will be addressed in future studies.

Reason for 8-week-old mice

Eight-week-old mice were selected for their fully developed immune systems and stable skin structure, ensuring a reliable model for evaluating skin healing. Their size and adaptability make surgical procedures smoother, facilitating consistent observations. Furthermore, this age is a standard in experiments, enhancing reliability and comparability of results.

We have included this explanation in the revised manuscript (Lines 319-324, 425-426).

Comment #26. Line 297: MSCs are not tumorigenic (unlike iPS cells). Please cite an appropriate source to clarify and support this statement.

Response: Thank you for your helpful comments. We have revised the expression to make it more rigorous and precise. (Lines 353-357)

Comment #27. Materials and Methods, Line 331: “(100 μL/well) and different concentrations of MSC-ext/PBS (100 μL/well)." Were the concentrations properly prepared? Adding 100 μL of extract to 100 μL of culture medium dilutes the protein concentration by half (!). Are the concentrations reported in the “Results” section accurate? What was the initial concentration of MSC-ext?

Response: We apologize for our insufficient explanation. The concentrations shown in Figure 2A represent the final concentrations. As you noted, the solution was diluted by half during preparation. Quantified by BCA analysis, the initial concentration of MSC-ext was 1880.7 µg/mL (average) per extraction. We have added the yield of MSC-ext and clarification in the Materials and Methods section. (Lines 76-77, 396)

Comment #28. Line 338: "per well at a concentration of 1 × 10⁵ cells/mL” – Media were added to the well, which would have altered the final cell concentration (number of cells/mL). Please clarify or provide the total number of cells per well. “(350 μg/mL concentration)” – Please specify the final concentration, taking into account the total volume of fluid in the well. Line 340: “200 μL of MSC-ext (350 μg/mL concentration)” – Was this the final concentration? Please verify concentration issues throughout the manuscript (not all instances are highlighted in this review), including lines 386-387 and others.

Response: We apologize for the lack of clarity in our explanation. The final concentration of cells was 2 × 104 cells/mL. In transwell assay, the setup consists of upper and lower chambers separated by a membrane. After adding a cell suspension (1 × 10⁵ cells/mL) to the upper chamber, MSC-ext or PBS were added to the lower chamber. Since the cells do not migrate to the lower chamber immediately, it is generally not appropriate to rely solely on the final cell concentration for evaluation. The liquid volume in the lower chamber exceeds the height of the upper chamber. We have added a diagram below.

Regarding the additive concentration issue: Thank you for your suggestion. We have revised the manuscript to address concerns about concentrations, and the Materials and Methods section now specifies the final concentrations (Lines 402-407, 418-419, 449-452).

Comment #29. Line 372: "In some experiments, the mice were euthanized on postoperative day 5." How many experiments were conducted in total? In Figure 4B, all time points (days 0-13) are shown for all 6 mice in one experiment. This is inconsistent with the text. Please revise or clarify this discrepancy.

Response: We apologize for the lack of clarity in our explanation. To clarify, the photographs were taken continuously over the course of the study. Therefore, the animals were not sacrificed on each day corresponding to the images.

For the 0-13 days observations, we used 6 animals (12 defect sites), while for the tissue section analysis, we used 4 animals (8 defect sites). (Lines 437-439)

Thank you so much for your constructive and precious comments, which raised the quality of our manuscripts.

Additional revisions:

We have some mistakes in the previous manuscript: (1) We have confirmed that "UE7T-13" is the more standard expression (“UE7T13” was used before the change), and we have updated the manuscript to reflect this change. We have also revised other places where errors similar to those pointed out in point 13 were found (Lines 254-256). We sincerely apologize for these errors. (2) Following the suggestion in point 14, we have also adjusted the layout of Figure 3. In addition, we have unified the expression and changed “fig.” to “figure”.

Round 2

Reviewer 2 Report (New Reviewer)

Comments and Suggestions for Authors

I appreciate the changes made to the manuscript and I now support its acceptance for publication.

Author Response

Thank you very much for recognizing and affirming our work. We are grateful for the time and effort you invested in reviewing our manuscript. Your support motivates us to continue striving for excellence in our research.

Reviewer 3 Report (New Reviewer)

Comments and Suggestions for Authors

Thank you very much for all the revisions and explanations provided. I greatly appreciate that the Authors have addressed all my comments so thoroughly and comprehensively. In my opinion, the quality and clarity of the work have significantly improved. I am also very grateful for the additional clarifications provided outside the article. I wish you success in your further research work.
Regarding the rearrangement of Fig. 2 – I also greatly appreciate this correction. The comment actually referred to Fig. 3, but there was a typo. Thank you for addressing this issue despite my oversight in providing the numbering, for which I apologize.
Just one very minor comment regarding my previous comment number 26:

Comment #26. Line 297: MSCs are not tumorigenic (unlike iPS cells). Please cite an appropriate source to clarify and support this statement.

Response: Thank you for your helpful comments. We have revised the expression to make it more rigorous and precise. (Lines 353-357)

Please add just a short phrase, for example: It should be emphasized that mesenchymal stem cells (MSCs) themselves do not undergo tumorigenic transformation; however, the growth factors they secrete can stimulate tumor growth. MSCs are generally considered safe regarding tumorigenic transformation under standard culture conditions. Studies have shown that MSCs do not spontaneously transform into cancerous cells. However, MSCs can secrete growth factors (e.g., VEGF, FGF, TGF-β, HGF), cytokines, and chemokines that promote angiogenesis and proliferation of cancer cells within the tumor microenvironment. This effect depends on the context, tumor type, and the interactions between MSCs and tumor cells.

Author Response

Thank you very much for promptly addressing our response. We deeply appreciate your recognition and affirmation of our work. It is truly an honor to receive your valuable suggestions, which have significantly improved the quality of our manuscript.

Round 2

Comment #26. Line 297: MSCs are not tumorigenic (unlike iPS cells). Please cite an appropriate source to clarify and support this statement.

Response: Thank you for your helpful comments. We have revised the expression to make it more rigorous and precise. (Lines 353-357)

Comment. Please add just a short phrase, for example: It should be emphasized that mesenchymal stem cells (MSCs) themselves do not undergo tumorigenic transformation; however, the growth factors they secrete can stimulate tumor growth. MSCs are generally considered safe regarding tumorigenic transformation under standard culture conditions. Studies have shown that MSCs do not spontaneously transform into cancerous cells. However, MSCs can secrete growth factors (e.g., VEGF, FGF, TGF-β, HGF), cytokines, and chemokines that promote angiogenesis and proliferation of cancer cells within the tumor microenvironment. This effect depends on the context, tumor type, and the interactions between MSCs and tumor cells.

Response. In response to your valuable suggestions, we have carefully revised our manuscript and included additional discussion. We also have added additional references to support it. This new content aims to address the concerns raised and provide a more comprehensive analysis of the relevant topic. We sincerely appreciate your insightful feedback, which has helped us enhance the clarity and depth of our work. (Lines 353-359)

(Original position in References):

  1. Barkholt, L.; Flory, E.; Jekerle, V.; Lucas-Samuel, S.; Ahnert, P.; Bisset, L.; Büscher, D.; Fibbe, W.; Foussat, A.; Kwa, M., et al. Risk of tumorigenicity in mesenchymal stromal cell-based therapies--bridging scientific observations and regulatory view-points. Cytotherapy. 2013, 15, 753–759, doi:10.1016/j. jcyt.2013.03.005.
  2. Liang, W.; Chen, X.; Zhang, S.; Fang, J.; Chen, M.; Xu, Y.; Chen, X. Mesenchymal stem cells as a double-edged sword in tumor growth: focusing on MSC-derived cytokines. Cell. Mol. Biol. Lett. 2021, 26, 3, doi:10.1186/s11658-020-00246-5.

This manuscript is a resubmission of an earlier submission. The following is a list of the peer review reports and author responses from that submission.

Round 1

Reviewer 1 Report

Comments and Suggestions for Authors

I would like to inform you of my comments about the manuscript entitled " Mesenchymal stem cell extract promotes skin wound healing". This study confirms that MSC-ext can promote the proliferation and/or migration of fibroblasts, epithelial cells, and endothelial cells, thereby enhancing skin wound healing. However, this manuscript still exhibits certain insufficiencies that warrant careful revision by the authors.

1. In the in vivo model, why were the HR mice treated with medication only within the first 5 days after wound construction, rather than continuously medicated until the 13th day? Furthermore, why was local injection chosen as the mode of administration instead of applying the medication directly onto the wound surface?

2. The abbreviations "MSC: Mesenchymal stem cell, PBS: phosphate buffered saline." at the end of each figure legend can be omitted. These abbreviations have already been clearly annotated in the main text and do not require repeated emphasis here.

3. To my knowledge, the formula for calculating the wound healing rate in Figure 3C should be (healed area / total area * 100%), which should yield a gradually ascending curve.

4. During the wound healing process, inflammatory cells directly promote the proliferation and migration of fibroblasts and capillaries by secreting growth factors and cytokines, thereby accelerating the formation and thickening of granulation tissue. Since the author measured the number of inflammatory cells in Figure 4D, it is also necessary to measure the thickness of granulation tissue in Figure 4A.

5. The author may consider adding a set of positive controls to further evaluate the in vivo reparative effect of MSC-ext.

6. An outstanding academic paper not only presents experimental results but also provides comprehensive discussions on these findings. The author should take note of this aspect. To enhance the discussion in this work, other relevant studies (PMID: 37325498, PMID: 37323217, PMID: 38939866) can be cited and discussed.

Comments on the Quality of English Language

Part of the English expressions need to be improved.

Author Response

We sincerely appreciate the reviewer's detailed and constructive feedback. Below, we present our point-by-point response to each comment.

Comment #1

In the in vivo model, why were the HR mice treated with medication only within the first 5 days after wound construction, rather than continuously medicated until the 13th day? Furthermore, why was local injection chosen as the mode of administration instead of applying the medication directly onto the wound surface?

Response #1

Thank you for your comments regarding the in vivo experimental system we employed. Regarding the administration of MSC-ext only during the first 5 days after wound construction, this approach was based on our previous study on periodontal tissue regeneration using MSC-ext (manuscript in preparation). While more frequent injections could potentially enhance efficacy, we aimed to align with clinical practice by minimizing the duration of therapeutic interventions. Our decision was informed by prior studies on periodontal tissue regeneration that demonstrated the effectiveness of a 5-day administration period. Moreover, we chose subcutaneous injections around the wound site instead of applying MSC-ext directly onto the wound surface. This choice was made because MSC-ext, when dissolved in PBS, could disperse extensively if applied directly to the wound surface, resulting in a minimal local concentration. We acknowledge that our original manuscript did not provide sufficient explanation for these methods. In response, we have included the aforementioned details in the revised manuscript (Lines: 109-112, 335-337). Additionally, we have added statements clarifying that the method of administration used for skin defects in this experiment has not yet been optimized (Lines: 250-252).

Comment #2.

The abbreviations "MSC: Mesenchymal stem cell, PBS: phosphate buffered saline." at the end of each figure legend can be omitted. These abbreviations have already been clearly annotated in the main text and do not require repeated emphasis here.

Response #2

Thank you for the suggestion. We have removed the description of “MSC” and “PBS” from the figure legends in the revised manuscript.

Comment #3

To my knowledge, the formula for calculating the wound healing rate in Figure 3C should be (healed area / total area * 100%), which should yield a gradually ascending curve.

Response #3

Thank you for your accurate remark. The vertical axis label in Figure 3C was indeed “Deficiency closure rate,” not the “Healing rate.” We have revised Figure 3C in accordance with your comment and updated the label accordingly. We appreciate the reviewer's constructive comment.(Line 339-340)

Comment #4

During the wound healing process, inflammatory cells directly promote the proliferation and migration of fibroblasts and capillaries by secreting growth factors and cytokines, thereby accelerating the formation and thickening of granulation tissue. Since the author measured the number of inflammatory cells in Figure 4D, it is also necessary to measure the thickness of granulation tissue in Figure 4A.

Response #4

Thank you for your valuable comment. We agree that inflammatory cells and granulation tissue formation are key aspects of wound healing. Our data demonstrated no significant difference in the number of inflammatory cells between the control and experimental groups. Although we considered observing the granulation tissue formation, the boundaries of the formed granulation tissue were unclear, which could have led to inaccuracies. Instead, we measured dermal thickness and assessed the amount of fibrous connective tissue formation, which includes granulation tissue, as presented in Figure 4B. The results demonstrated that the dermis was significantly thicker in the MSC-ext group, suggesting enhanced connective tissue formation, including granulation tissue. As you noted, we have added this information to the revised manuscript (Line 124-128) to address the incomplete description of granulation tissue formation in the original submission.

Comment #5

The author may consider adding a set of positive controls to further evaluate the in vivo reparative effect of MSC-ext.

Response #5

We appreciate the reviewer’s valuable comment. We agree that adding a set of positive controls would provide a more comprehensive evaluation of the in vivo reparative effects of MSC-ext. To address this, we plan to conduct further experiments comparing MSC-ext with established treatment options such as bFGF, which is clinically used for refractory skin defects, as well as with MSC cell transplantation, MSC culture supernatants, and MSC-derived exosomes, which are currently in the preclinical stage. Additionally, since MSC-ext is rich in bFGF, a growth factor widely used in clinical skin wound treatment, we anticipate that MSC-ext’s effects may be comparable to those of bFGF. The purpose of this manuscript is to conduct an initial investigation into the potential therapeutic efficacy of MSC-ext. We acknowledge that the lack of comparison with other treatment modalities is a limitation of the current study and will address this point in future research. We added above limitation of this study in the discussion section (Line 253-256).

Comment #6

An outstanding academic paper not only presents experimental results but also provides comprehensive discussions on these findings. The author should take note of this aspect. To enhance the discussion in this work, other relevant studies (PMID: 37325498, PMID: 37323217, PMID: 38939866) can be cited and discussed.

Response #6

We greatly appreciate the reviewer’s constructive comments regarding the content of our discussion. We apologize for the lack of information related to skin defect treatment in our original manuscript. In response to your feedback, we have included the three suggested references (Ref #44-46 in revised manuscript) and expanded our discussion to cover new biomaterials and stem cells that are being used for skin defect treatment, along with emerging therapies currently in development (Lines: 258–264). We are grateful for your feedback, which has significantly improved the content of our paper.

Reviewer 2 Report

Comments and Suggestions for Authors

The authors researched the effects of MSC-ext on wound healing. The research is of interest. However, the data are too preliminary to get to their conclusion.

1. Figure 1, the authors only showed the WST-8 results, much more data are needed to demonstrate the cell proliferation of cells.

2. Figure 2, the authors only showed the transwell results, more data such as the expression of cell markers and wound healing experiments are needed to demonstrate the cell migration of cells.

3. Figure 3, why do you use 4-mm model? In mouse model, 6-mm or 10-mm model are usually used. How to apply the MSC-ext? The description should be in detail. Why use Hos:HR mice? Hair follicle stem cells are also important factors in wound healing.

4. How long can the molecules keep activities in the model? The data to validate the model are needed.

5. What are the main functional molecules in MSCC-ext? What is the potential mechanism? They did not show any data to answer these questions.

Comments on the Quality of English Language

Moderate editing of English language required.

Author Response

REVIEWER 2

We appreciate the reviewer’s thought-provoking comments. Below, we present our point-by-point response to each comment.

Comment #1

Figure 1, the authors only showed the WST-8 results, much more data are needed to demonstrate the cell proliferation of cells.

Response #1

Thank you for your comment. We understand the concern about relying solely on the WST-8 assay results to assess cell proliferation. There are several methods to test cell proliferation, including BrdU incorporation, Ki-67 staining, and flow cytometry. The WST-8 assay was chosen for its ability to provide a quick and reliable estimation of cell viability and proliferation based on metabolic activity. Although it is an indirect measure, it is widely used and well-validated in similar studies, making it a suitable method for initial screening. In our previous study by Peng et al., Ki-67 assays were conducted to confirm that MSC-ext significantly enhanced periodontal fibroblasts proliferation (Peng, Y.; Iwasaki, K.; Taguchi, Y.; Umeda, M. The Extracts of Mesenchymal Stem Cells Induce the Proliferation of Periodontal Ligament Cells. Journal of Osaka Dental University 2023, 57, 119–124, doi:10.18905/jodu.57.1_119.). In our current study, we carefully interpreted the results within the context of these limitations, presenting the findings as preliminary evidence rather than as definitive proof of proliferation. We recognize the need for more comprehensive data and plan to include additional experiments in future studies to provide a more thorough analysis of cell proliferation.

Comment #2

Figure 2, the authors only showed the transwell results, more data such as the expression of cell markers and wound healing experiments are needed to demonstrate the cell migration of cells.

Response #2

Thank you for your insightful feedback. We appreciate the suggestion to include additional data, such as cell markers expression and wound healing assays, to better demonstrate cell migration. Due to certain constraints, we focused on the transwell migration assay, a widely accepted method for assessing cell migration.

Actually, in this experiment, we consider that in the positive group, MSC-ext not only promotes cell wandering quantitatively, but also promotes cell stretching. Actin proteins, closely associated with cell migration, promote the formation of filopodia that drive and coordinate cell migration. [1-2]

(Please refer to Images for this explanation on the response letter file)

As shown in the figure, in the positive group, MSC-ext promotes cell stretching. The arrows point to the filopodia of the cell.

However, we recognize the value of incorporating additional experiments to provide a more comprehensive analysis. We acknowledge that the transwell assay alone may not fully capture the complexity of the cell migration process. In the revised manuscript, we have carefully discussed these limitations and interpreted the results as part of a broader investigation of cell migration.

We are currently preparing an article that provides a detailed investigation of the effects of MSC-ext on cell proliferation and migration. Consequently, in this study, we focused on wound healing experiments to highlight the migratory effects. While this study focused on the transwell assay results, we acknowledge the importance of a more comprehensive approach. We will prioritize these additional experiments in our future research to provide a more detailed understanding of the cell migration process.

References:

[1] Fritz-Laylin LK, Titus MA. The evolution and diversity of actin-dependent cell migration. Mol Biol Cell. 2023;34(12):pe6. doi:10.1091/mbc.E22-08-0358

[2] Bodor DL, Pönisch W, Endres RG, Paluch EK. Of Cell Shapes and Motion: The Physical Basis of Animal Cell Migration. Dev Cell. 2020;52(5):550-562. doi:10.1016/j.devcel.2020.02.013

Comment #3

Figure 3, why do you use 4-mm model? In mouse model, 6-mm or 10-mm model are usually used. How to apply the MSC-ext? The description should be in detail. Why use Hos:HR mice? Hair follicle stem cells are also important factors in wound healing.

Response #3

Thank you for your question regarding the experimental design shown in Figure 3. We appreciate the opportunity to clarify our choices.

Reason for the 4-mm Model: We selected the 4-mm wound model because of its suitability for our study. This wound size provides a manageable and consistent model, allowing for the precise assessment of wound healing and the localized effects of MSC-ext. The 4-mm size ensures reproducibility across experiments and facilitates detailed histological analysis, which can be more challenging with larger wounds.

Application of MSC-ext: MSC-ext was stored on ice prior to injection to prevent protein inactivation. Injections were administered subcutaneously at four points (0°, 90°, 180°, and 270°) around the circular wound, with 50 µL per point (200 µL per wound). The application was performed under sterile conditions and repeated every 24 hours for 5 days to maintain MSC-ext levels. These information was added in revised manuscript. (Line 335-337)

Use of Hos-HR mice: Hos-HR mice were selected for this study due to their unique characteristics, particularly their hairless phenotype, which allows for the direct observation of wound healing without the confounding effects of hair regrowth.

We initially started a pilot study in C57BL/6 mice but switched to Hos-HR mice in the first trial because hair growth in C57BL/6 mice made wound observation unclear and required considering the hair cycle during administration. Although hair follicle stem cells play a crucial role in wound healing, our study focused on isolating and evaluating the effects of MSC-ext in a controlled environment. The absence of hair in Hos-HR mice allowed for a more precise assessment of wound closure and tissue regeneration. We acknowledge the role of hair follicle stem cells in wound healing. For this initial study on MSC-ext in skin, the Hos-HR mouse model was chosen because of its suitability for achieving more accurate and reproducible results. The original manuscript did not adequately explain the rationale for selecting this animal model. Therefore, we have included this explanation in the revised manuscript (Lines 328–329). Thank you for bringing this to our attention.

Comment #4

How long can the molecules keep activities in the model? The data to validate the model are needed.

Response #4

Thank you for raising this important question. We acknowledge that the duration of MSC-ext activity within the model is a critical factor that has not been determined in our current study. In this study, we administered MSC-ext without any carrier materials. If MSC-ext had not shown efficacy in wound healing, I planned to use carrier materials. However, our results demonstrated that MSC-ext effectively reduced skin defects after 5 days of continuous administration, leading us to proceed with this method in our study. The concern about how long MSC-ext remains in the local area after injection is indeed an important point. We recognized that MSC-ext likely dispersed relatively quickly from the injection site. To address this, we administered MSC-ext via five consecutive daily injections to maintain local concentrations. We acknowledge that this approach has not yet been optimized, which is a limitation of our study. We have revised the manuscript to emphasize this point (Lines 250–252). Thank you for highlighting this crucial issue.

Comment #5

What are the main functional molecules in MSC-ext? What is the potential mechanism? They did not show any data to answer these questions.

Response #5

Thank you for your thoughtful comment regarding the main functional molecules in MSC-ext and their potential mechanisms of action. We recognize the importance of identifying and understanding the functional molecules within the MSC-ext and their underlying mechanisms of action. Although our current study focused primarily on evaluating the regenerative effects of MSC-ext, we did not specifically identify the molecular components responsible for its wound healing activity. However, based on our previous research and existing literature, we can provide an overview of the main functional molecules and their potential mechanisms.

MSC-ext is recovered from cultured cells through freeze-thaw cycles, centrifugation, and filtration and is believed to contain a wide array of cell-derived factors. In our previous study, LC-MS/MS analysis revealed that MSC-ext contains 4,388 protein components. Due to the extensive number of factors involved, pinpointing the most critical ones has been challenging. However, our prior research indicated that MSC-ext is particularly rich in growth factors such as bFGF and HGF, which are known to promote wound healing. We believe that these growth factors may have contributed to the observed effects on skin defects in this study. Additionally, the in vivo data demonstrated that MSC-ext significantly thickened the dermis and increased the number of Ki-67-positive fibroblasts, suggesting that fibroblast proliferation induced by MSC-ext may play a crucial role in promoting wound healing. We added above descriptions in the revised manuscript (Line 185-190). We thank the reviewer for their important comments.  

Reviewer 3 Report

Comments and Suggestions for Authors

Dear Authors,

The current manuscript aims to provide a clinically relevant treatment for wound healing. However, I have many concerns/suggestions for the current study.

1. The introduction of the paper is sufficient but it lacks the proper references relevant to the MSC secreted factors or extracts in wound healing. There are many reports available regarding this, however, I come across this one (PMID: 31761327). This particular report explained how human mesenchymal stromal cells conditioned media reverses low serum and hypoxia-induced inhibition of wound closure.

2. The Materials and Methods section lacks all the experimental details such as in vitro experiments details for extract media volume, and animal experiment details for treatment application and location. In addition, no details are available for the location of histological analysis.

3.  Results are poorly written and it looks like materials and methods. There are no details of secreted extract components that are needed to back up this data. There is no way authors can explain this finding and different responses with different cells. Figure 1: My only concern in this figure is that the authors did not explain how they used the extract in 96-well plate. E.g. how much volume for each concentration used? Similarly, the same issue with the migration assay and no data was provided for migrated cells. Only 5-day histology was provided and there is no way the wound closed in this duration as shown in the histology images. There is no end-point data of histology included in this manuscript. Wound healing curve analysis using a t-test is wrong when you have a time-line study.

4. The discussion is poorly written and does not match the data presented in the manuscript.

Overall, this study needs serious work before publication. 

Comments on the Quality of English Language

ok

Author Response

REVIEWER 3

We thank the reviewer for many constructive comments.

Below, we present our point-by-point response to each comment. We believe that the revised paper has been significantly improved by the reviewer’s inputs.

Comment #1

The introduction of the paper is sufficient but it lacks the proper references relevant to the MSC secreted factors or extracts in wound healing. There are many reports available regarding this, however, I come across this one (PMID: 31761327). This particular report explained how human mesenchymal stromal cells conditioned media reverses low serum and hypoxia-induced inhibition of wound closure.

Response #1

We appreciate the reviewer’s constructive comment. We sincerely apologize for the inadequate description in the background section. The paper (PMID: 31761327) provided by the reviewer demonstrates how MSC-derived conditioned media can reverse the inhibition of wound closure caused by low serum and hypoxia, emphasizing the role of MSC-secreted factors in wound healing. In response to the reviewer’s suggestion, we have added references and descriptions regarding MSC-derived factors in the revised manuscript (References 19-23)(Line 48-50).

Comment #2

The Materials and Methods section lacks all the experimental details such as in vitro experiments details for extract media volume, and animal experiment details for treatment application and location. In addition, no details are available for the location of histological analysis.

Response #2

Thank you for highlighting the importance of describing the experimental conditions. We have added the requested information on the experimental details, as noted by the reviewer (Lines 300, 308-309, 317).

Comment #3

Results are poorly written and it looks like materials and methods. There are no details of secreted extract components that are needed to back up this data. There is no way authors can explain this finding and different responses with different cells. Figure 1: My only concern in this figure is that the authors did not explain how they used the extract in 96-well plate. E.g. how much volume for each concentration used? Similarly, the same issue with the migration assay and no data was provided for migrated cells. Only 5-day histology was provided and there is no way the wound closed in this duration as shown in the histology images. There is no end-point data of histology included in this manuscript. Wound healing curve analysis using a t-test is wrong when you have a time-line study.

Response #3

We sincerely appreciate the reviewer’s constructive feedback and apologize for the lack of a detailed description of the experimental procedures.

In response to the reviewer's suggestion, we have added information on the contents of MSC-ext and detailed the experimental conditions (Line 308-309). For animal experiments, we focused on histological images taken 5 days after MSC-ext administration. This time point was selected because it marks the period when a significant difference in wound healing between the control and experimental groups begins to emerge, allowing us to observe histological changes related to the mechanism of MSC-ext. Experiments were conducted in healthy, relatively young mice, where normal wound healing was anticipated. The primary objective was to compare wound closure speeds, making the 5-day data particularly critical for our analysis. In future studies, we plan to apply MSC-ext to models of intractable skin defects such as diabetes mellitus, and will include endpoint data analysis in these experiments. We appreciate the reviewer’s comments regarding the statistical analysis of the data. For the time course of wound healing shown in Figure 3C, we used the Student's t-test to compare the control and experimental groups at each specific time point. Although data were recorded continuously over time, statistical comparisons were performed at each specific time point. This approach is consistent with similar studies (e.g., PMID: 27466403, PMID: 31028245, PMID: 37086260). We hope this clarifies our methodology.

Comment #4

The discussion is poorly written and does not match the data presented in the manuscript.

Response #4

We apologize for the inadequacies in the discussion section. We have revised it to better reflect the data presented in the paper, as suggested by the reviewer, including description of the contents of MSC-ext and a discussion of the limitations of the study. Specifically, we also added a new discussion on the effects of MSC-ext on cell migration and the number of inflammatory cells in vivo, which was not addressed in the original manuscript (Lines 185-190, 201-208, 237-242). We sincerely thank the reviewer for their constructive comments.

Round 2

Reviewer 1 Report

Comments and Suggestions for Authors

The author has revised the manuscript according to my suggested opinions. I recommend acceptance.

Reviewer 2 Report

Comments and Suggestions for Authors

The authors did not add any data to address the concerns.

Comments on the Quality of English Language

Moderate editing of English language required.